# Integrated bulk RNA and single-cell analysis with experimental validation reveal oxidative stress-related diagnostic biomarkers for osteoporosis

Cheng Zhong[1]*, Liping Zhong[2]

1 Department of Orthopedics, Jiangmen Hospital of Traditional Chinese Medicine Affiliated to Jinan University, Jiangmen, China, 2 Department of Cardiothoracic surgery, Jiangmen Hospital of Traditional Chinese Medicine Affiliated to Jinan University, Jiangmen, China

* 20216020037@stu.gzucm.edu.cn

## Abstract

Osteoporosis (OP) is a systemic skeletal disorder characterized by reduced bone mass and deterioration of bone microarchitecture, which increases fracture risk and impairs physical function. This study explores the role of CHRM2 in osteogenic differentiation and evaluates its potential as a biomarker for OP. Single-cell RNA sequencing revealed distinct differences in cell type distributions between OP patients and healthy controls, notably an increase in M1 macrophages and regulatory T cells in OP patients. Functional enrichment analysis underscored the involvement of regulatory T cells in OP pathogenesis. Furthermore, CHRM2 was identified as a key gene associated with oxidative stress. In vitro experiments demonstrated that CHRM2 knockdown enhanced osteogenic differentiation while suppressing cell proliferation, likely via interactions with COL4A2. These findings suggest that CHRM2 plays a negative regulatory role in osteogenic differentiation and may serve as both a diagnostic biomarker and a potential therapeutic target for early-stage OP.

## Introduction

With an aging population, osteoporosis (OP) had become a growing public health challenge worldwide [1]. Although the pathogenesis of OP has been extensively studied, the exact mechanism is still not fully understood due to the complex etiology [2]. Therefore, it is very vital to identify effective biomarkers for the early diagnosis, prevention, and treatment of OP. More importantly, early identification of OP patients and effective intervention at an early stage can fundamentally prevent severe complication, such as fractures [3–5]. As a result, the application of bioinformatics is particularly crucial, as it enables the precise identification of OP-related biomarkers, facilitating accurate risk prediction and providing a foundation for developing individualized treatment regimens for OP patients.

**Data availability statement:** The datasets analyzed in this study are publicly available from the Gene Expression Omnibus (GEO) database (https://www.ncbi.nlm.nih.gov/geo/). Specifically, bulk RNA-seq and single-cell RNA-seq (scRNA-seq) transcriptome data were obtained from the following GEO accession numbers: GSE56116, GSE62402, GSE147287, and GSE169396. For downstream analyses, we selected an osteoporotic (OP) sample from GSE147287 and a normal bone tissue sample from GSE169396. Datasets GSE56116 and GSE62402 provided essential clinical information, including patient age, gender, and disease duration, which were used to support further analysis. In addition, a list of oxidative stress-related genes (n = 2,486) with a relevance score ≥ 0.4 was retrieved from the GeneCards database (https://www.genecards.org) by using "oxidative stress" as the search term. All datasets used are freely accessible through the provided GEO accession numbers and links.

**Funding:** This study was supported by the National Natural Science Foundation of China (Grant No. 82171372), Science and Technology Project of Jiangmen City(2021YL03002). The funders had no role in study design, data collection and analysis, decision to publish, or preparation of the manuscript

**Competing interests:** The authors have declared that no competing interests exist.

Currently, the gold standard in clinical diagnosis of OP is the measure of bone mineral density by dual emission X-ray absorptiometry (DXA) [6]. This method assesses fracture risk by measuring bone mineral density in the femoral and lumbar spine. However, the measurement results can be disturbed by positioning errors, soft tissue, and graft artifacts. Additionally, osteophytes, ankylosing spondylitis, lymphoma, and cancer bone metastases caused by osteoarthritis may also affect its accuracy [7]. In recent years, biomarkers developed to optimize the clinical management of OP have shown high sensitivity and reliability in the detection of people at high risk of fracture [8,9]. These biomarkers not only help predict the risk of OP, but can also be utilized to identify potential therapeutic targets and explore the associated pathologic mechanisms [10]. Moreover, the role of the immune system in orthopedic diseases was confirmed, promoting the development of the field of "bone immunology" [11,12].

Bioinformatics, as a discipline combining biotechnology and information technology, is devoted to the storage, retrieval and analysis of biological data, which is widely used in the fields of medicine, life science and bioengineering [13]. In this study, we conducted comprehensive bioinformatic analyses to integrate RNA-seq and single-cell transcriptome data alongside clinicopathological information from OP patients. We identified differentially expressed genes (DEGs), performed functional enrichment analyses, and pinpointed CHRM2 as the optimal core gene. The identification of CHRM2 highlights its potential as a biomarker, particularly for the early diagnosis of OP, with promising clinical applications. To further investigate the role of CHRM2, we conducted a series of key experiments, including cell-based assays and gene knockout studies, to validate its function. These experiments elucidated the molecular mechanisms of CHRM2 in OP, providing robust evidence to support its use as a biomarker for early diagnosis and risk prediction. Additionally, our findings pave the way for the development of personalized treatment strategies tailored to OP patients.

## Methods

### Public data collection

The original bulk RNA-seq and single-cell RNA-seq (scRNA-seq) transcriptome data were obtained from the Gene Expression Omnibus (GEO) database (https://www.ncbi.nlm.nih.gov/geo/), specifically from datasets GSE56116, GSE62402, GSE147287, and GSE169396. For further analysis, we selected an OP sample from GSE147287 and a normal bone tissue sample from GSE169396. The GSE56116 and GSE62402 datasets provided detailed clinical information, including patient age, gender, and disease course, which served as critical foundational data for subsequent analyses. Additionally, we queried the GeneCards database (https://www.genecards.org) using "oxidative stress" as a keyword and identified 2,486 oxidative stress-related genes with a relevance score ≥ 0.4 [14].

### Processing of scRNA-seq data

The quality filtering of scRNA data was conducted with multiple filtering parameters including >5% of mitochondrial genes, cells expressing the lower number of genes (<200 or > 2500 genes), and genes only uniquely expressed in <3 cells [15].

Moerover, we removed the potential doublets using the DoubletFinder package (version 2.0.3) of the R [16]. Then, the scRNA count data was normalized using the Log-Normalize algorithm in Seurat (v4.0.4) package. The top 2000 highly variable genes (HVGs) were identified, centered, and scaled from the normalized expression matrix before we performed the principal component analysis (PCA) based on these HVGs [17]. The batch effects were removed by the Harmony package (version 1.0) of R based on the top 50 PCA components.

For dimensional reduction, t-stochastic neighboring embedding (t-SNE) and Uniform Manifold Approximation and Projection (UMAP) analysis methods were performed on the HVGs. Unsupervised clustering of cells was carried out using the FindClusters command with a resolution of 1. Cells with similar transcriptome profiles clustered together, and the clusters were subsequently annotated to different cell types based on the expression of specific well-established cell markers.

To annotate the cell clusters, DEGs with high discrimination abilities between the custers were identified with the FindAllMarkers function in Seurat using the default non-parametric Wilcoxon test with Bonferroni correction. The cell groups were annotated based on the DEGs and the well-known cellular markers from the literature.

## DEGs identification and enrichment analysis

We identified DEGs within specific clusters compared to other clusters using the "FindMarkers" function in Seurat, based on the Wilcoxon test (adjusted P-value < 0.05 and logfc.threshold = 0.25). To determine cluster-specific overrepresented Gene Ontology (GO) biological processes, we utilized the compareCluster function from the clusterProfiler package (version 3.14.3) in R. For bulk RNA-seq data, DEGs between normal and OP tissues were identified using the limma R package, applying thresholds of adjusted P-value < 0.05 and |log2Fold Change (FC)| > 0.585 for downstream analyses. Subsequently, GO and KEGG functional enrichment analyses were performed using the clusterProfiler and enrichplot R packages to identify enriched signaling pathways associated with OP.

## Trajectory analysis of single cells and cell–cell communication

The single-cell pseudo-time trajectories were performed by the monocle2 package (v2.8.0) [18]. The scRNA data counts extracted from the Seurat data were served as the inputs in the newCellDataSet function to create an object with the parameter expressionFamily = negbinomial.size [18]. Only genes with the mean expression ≥ 0.1 were utilized in the trajectory analysis. The "plot cell trajectory" function applied to order and visualize the cells. The cell–cell communications were analyzed using CellChat package in R software. The human database of CellChatDB.human, including interactions from 'secreted signaling', 'ECM-receptor' and 'cell–cell contact', was utilized for further analysis. The standard protocol of CellChat was applied to normalized scRNA-seq counts using the following functions "dentifyOverExpressedGenes" "identifyOverExpressedInteractions", and "computeCommunProb", "filterCommunication".

## Cell lines and culture

The HUM-iCELL-s011cell line was cultured in minimum essential medium (MEM, 6123020)-α containing 10% fetal bovine serum(10099–141) (all from Procell Life Sci-ence and Technology Co., Ltd.) and 1% penicillin–streptomycin(P1400) at 37°C in an atmosphere of 5% CO2

## Cell transfection

The small interfering RNAs (siRNAs) used in experiments were purchased from OBiO Technology Co., Ltd. Cells were seeded in six-well plates and grown to 80%-90% confluence. The siRNAs (siR-CHRM2, siR-COL4A2, or the negative control) were mixed with Opti- MEM (Gibco) for 5 min to conFig the plasmid to a working concentration of 50nM. Lipofect-amine 2000 Transfection Reagent (Invitrogen, 11668–019) was mixed with Opti-MEM for 5 min. The two mixtures were then combined together for 15 min and added to every well.

## RNA fluorescence in situ hybridization (FISH) and Immunofluorescence (IF)

The FISH kit (Servicebio, Wuhan, China) was performed based on the manufacturer's instructions, and a fluorescence microscope was used for visualization.

IF was detected using a fluorescence microscope to identify protein localization and expression. Cells were fixed with 2ml 4% paraformaldehyde for 15min, and then permeabilized in 0.5% Triton X-100 for 10min at room temperature, followed by blocking with goat serum for 30min. Then, the cells were incubated with the primary antibody at 4°C overnight. After washing with PBS, the cells were incubated with the secondary antibody at 37°C for 50min, then treated with 4′,6-diamidino-2-phenylindole (DAPI, C1005, Solarbio, China) for 10min in the dark condition to stain the nuclei. The cells were visualized using a fluorescence microscope after washing with PBS(G4202, Servicebio, China).

## Real-time reverse transcription-quantitative polymerase chain reaction (RT-qPCR) assay

The RNAE×ZOL (ECOTOP, China) was utilized to extract total cellular RNA according to the protocol. RNA was reverse transcribed to cDNA by the PrimeScript RT reagent kit (BL699A, EZBioscience, China). EZBioscience 2 × SYBR Green qPCR Master Mix (A301-10, GenStar, China) conducted the procedure. GAPDH was chosen as internal reference. Expression levels of mRNAs were measured as $2^{-\Delta\Delta CT}$. Primers for mRNAs were provided by TSINGKE (Beijing TSINGKE Biotech Co., Ltd., China) and shown.

OPN Forward: CATCACCTGTGCCATACCAG Reserve: CTCATGGCTTTCGTTGGACT
RUNX2 Forward: GAGTGGACGAGGCAAGAGTT Reserve: GAGGCGGTCAGAGAACAAAAC
CHRM2 Forward: AAGCGGACCACAAAAATGGC Reserve: ATCTTTGGAATGGCCCAGGG
GAPDH Forward: AGAAGGCTGGGGCTCATTTG Reserve: AGGGGCCATCCACAGTCTTC

## Western blot

Cells lysis was performed utilizing RIPA buffer(R0010-100ml, Solarbio, China), supplemented with a protease inhibitor cocktail (CoWin Biosciences, China). Protein concentrations were determined by the bicinchonininc acid (BCA, ECOTOP, China). The membranes were blocked in protein-free rapid blocking buffer (PS108, Epizyme) for 0.5h at 24°C, and incubated with first antibodies at 4°C overnight. Then, the proteins were transferred into polyvinylidene fluoride membranes. The membranes were incubated with the secondary antibody at 24°C for 1.5h. The protein signals were detected using the enhanced chemiluminescence reagent (Beyotime, Shanghai, China) on a Tanon 5200 chemiluminescent imaging system (Tanon, Shanghai, China) and band intensities were measured using ImageJ.

## Alkaline phosphatase (ALP) staining and activity assays and Alizarin red staining

The differentiated cells were washed with PBS, fixed with 4% paraformaldehyde, and stained using a 5-bromo- 4-chloro-3-indolyl phosphate (BCIP)/nitro blue tetrazolium (NBT) ALP Color Development Kit (Beyotime) based on the instructions. The ALP activity kit (E1043, Elabscince) was utilized for ALP activity assays. As for Alizarin red staining, the cells were visualized in a microscope. Then, a 10% hexadecylpyridinium chloride monohydrate (B127, Solarbio) solution was added to dissolve the alizarin red bound to mineralized nodules. The optical density of the resultant solution was analyzed based on a microplate reader (Biotek).

## Cell proliferation and flow cytometry assays

The cell viability was cell detected by the counting kit-8 (CCK8, E20221116T06,ECOTOP, China) assay. In brief, cells were seeded in 96-well plates ($5 \times 10^4$ cells/well) with 100ul cell suspension, and six replicate wells were assigned to each treatment group. Cells were cultured under standard conditions, then 10μl CKK8 reagent was added, followed after 2–3 using a microplate reader at 450nm. As for the cell cycle, flow cytometry was performed by a detection kit (FXP0211, Elabscience), using the FACSCalibur BD flow cytometer, according to the manufacturer's instructions.

## Quantification and statistical analysis

Each experiment was repeated at least three times. Differences between experimental groups were assessed using Student t test, Wilcoxon test or one-way ANOVA. The quantitative data were presented as mean ± standard error of the mean (SEM). All tests are bilateral, and P value < 0.05 is considered significant. Statistical analyses were performed using R (version 4.3.1) or GraphPad Prism 8.0 or SPSS software.

## Results

### Single cell analysis reveals the heterogeneity of gene expression between OP and normal bone tissue

As shown in Fig 1A, 1B, and 1C, the number distribution of genes and the proportion distribution of mitochondrial genes in all cells were discovered, which showing that the data distribution after quality control is relatively uniform and meets the expected standards. The analysis of the average expression and variance of all genes identified 2000 HVGs (Fig 1E). The hypervariable genes, which differ significantly between normal bone tissue and osteoporosis patients, will be used for subsequent reduction and cluster analysis. The PCA showed that the top 30 principal components explained most of the variation in the data. Fig 1D shows the standard deviation of each principal component, with the first 30 principal components selected for downstream analysis. In the dimensionality reduction of UMAP, cells from normal bone tissue and OP patients show a distinct separation, with different colors indicating different cell populations (Fig 1F and 1G). It was revealed that FABP4, SPP1, TM4SF1, NTS and other genes were significantly up-regulated in the cells of OP patients, suggesting that these genes may play an important role in the occurrence and development of OP; meanwhile, CCDC85, HSPA1A, KRT19 and other genes were significantly up-regulated in normal bone tissue cells, which may play an important role in maintaining the function and structure of normal bone cells (Fig 1E). The results of DEGs analysis further indicated that some genes play an important role in the occurrence and development of osteoporosis. These findings provide new clues for the study of the mechanism of osteoporosis and the exploration of therapeutic targets.

### Cellular constitution of OP lesions

In the cluster analysis, we performed detailed resolution optimization of single-cell RNA sequencing data, and finally selected parameters with a resolution of 1.0 for cluster analysis (S1 Fig). Unbiased clustering of the cells identified 9 main cell clusters, based on t-SNE and UMAP analyses according to their gene profiles and canonical markers (Fig 2A, 2B). In particular, they were as follows (S2 Fig ): (1) the monocytes highly expressing CD14; (2) M1 macrophages highly expressing FCGR3B and FCGR1A; (3) M2 macrophages characterized with high MSR1, CD163, MRC1, and CSF1R expression; (4) the CD8$^+$ T cells specifically express the markers GZMK, CD8A, CD8B, and CD3D; (5) the CD4$^+$ memory T cells with high expression of IL7R, CCR7, and CD27; (6) the B cells specifically expressing CD37 and CD79A; (7) the regulatory T cells expressing FOXP3, LAG3, NRP1, and ITGA2; (8) the NK cells highly expressing CD160, CD247, CL3, GZMB, NKG7, and GNLY; (9) the fibroblasts expressing FGF7 and MME. There were significant differences in the distribution of different cell types in osteoporosis patients and normal bone tissue. Through the cell annotation, it was found that the proportion of M1 macrophages and regulatory T cells increased significantly in the OP patients, while the proportion of CD4$^+$ memory T cells and fibroblasts was higher in normal bone tissue. These differences may reflect the pathological changes and immune responses in osteoporosis.

### Trajectory of OP lesions

Using Monocle2, we performed single-cell pseudotime analysis to infer cell trajectories and examine the dynamic changes in cell populations during disease progression. The distribution of different cell types along the pseudotime trajectory and their progression over time are illustrated in Fig 2C and 2D. The pseudotime analysis revealed that osteoporosis-associated

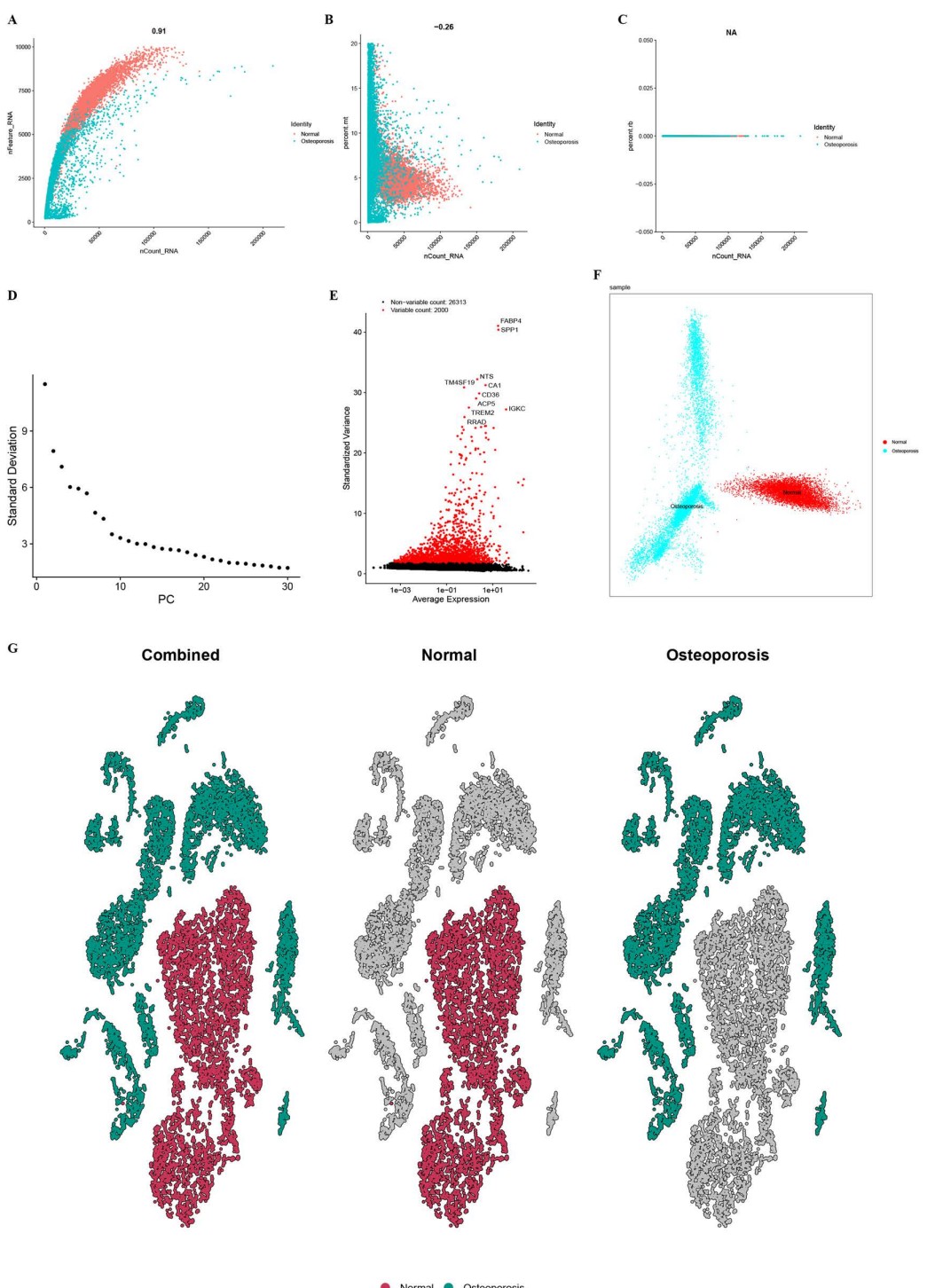

**Fig 1. Single cell transcriptome analysis showed differences between normal and osteoporosis samples.** (A) The relationship between nCount RNA and nFeature RNA of the two groups of samples. (B) The relationship between nCount_RNA and mitochondrial gene ratio in the two groups of samples. (C) The relationship between nCount RNA and ribosome gene ratio of the two groups of samples. (D) Standard deviation distribution in PCA. (E) The relationship between average and standard variance of gene expression. (F) The distribution of samples in a two-dimensional PCA plot. (G) Cell population distribution shown by PCA dimensionality reduction of all samples, normal sample, and the osteoporosis sample.

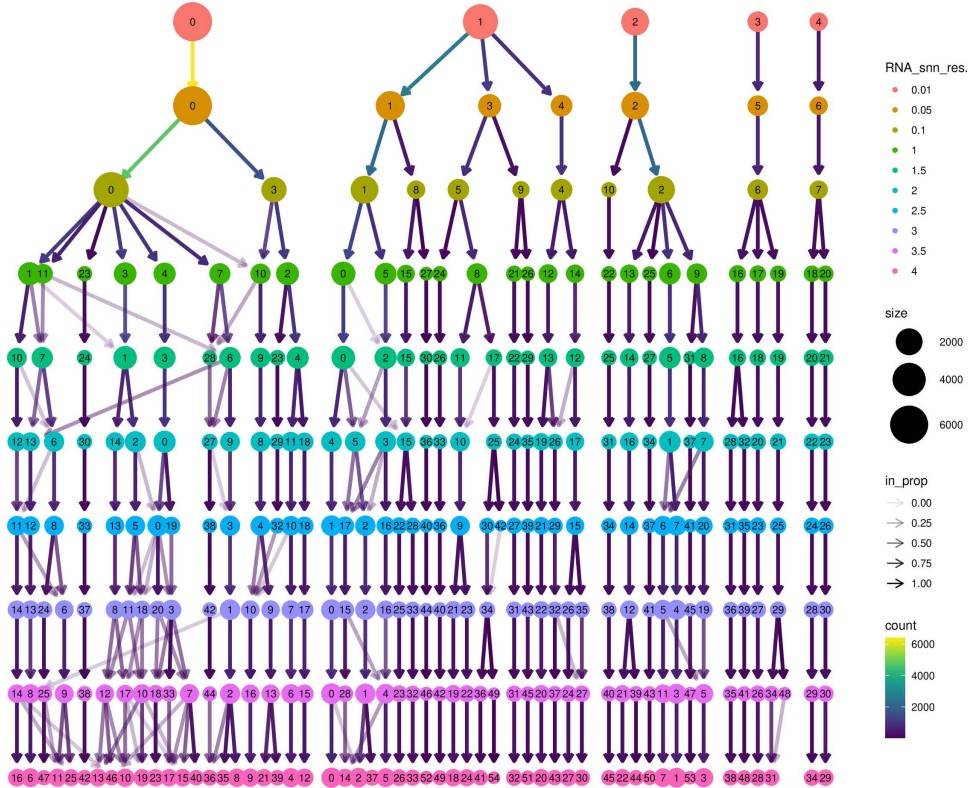

**Fig 2. Cell type clustering and trajectory analysis of scRNA data.** (A) Cell types distribution shown by t-SNE dimensionality reduction. (B) Cell types distribution as shown by UMAP dimensionality reduction. (C) Two-dimensional projections of Pseudo-time analysis of different cell types. (D) Pseudo-time analysis results. (E) The gene function classification.

cell types, such as M1 macrophages and regulatory T cells, followed distinct evolutionary paths along the pseudotime continuum, suggesting a close association with disease progression. Notably, regulatory T cells were predominantly concentrated in the early stages of disease development, whereas fibroblasts were more abundant in the later stages. These findings provide insight into the temporal involvement of specific cell types in OP pathogenesis.

## Cell–cell communication to construct the communication networks

Cell-cell communication analysis included Secreted Signaling, ECM-Receptor Interaction, Cell-cell Contact, which accounted for 61.8%, 21.7%, and 16.5%, respectively. Further analysis revealed that 47.9% of cell communication was accounted for by Heterodimers, which mainly involved complexes formed by different kinds of receptor subunits with unique functions in signaling. Others accounted for 52.1%, including cellular communication mechanisms not classified in the above categories. Functional analysis discovered that 27% of the genes involved in cell communication are involved in known signaling pathways in the KEGG database, and 73% of the genes have been reported in the existing literature, which provides us with a large amount of information on the function of these genes and their role in cell communication (Fig 2E). In summary, cell communication exhibits complex and diverse interactions between different types of cells. These findings not only reveal the diversity and complexity of cellular communication, but also provide insights into the finer aspects of specific signaling pathways

As shown in S3A and S3B Fig, regulatory T Cells have a significant centrality in the network, indicating an important role in intercellular communication, which not only have strong communication with various cell types such as M2

macrophages, monocytes and fibroblasts, but also play a key role in the regulation of immune response. S3C Fig showed the position of regulatory T cells in the cellular communication network. Regulatory T cells have strong communication with a variety of other cell types, such as monocytes, M1 macrophages, M2 macrophages, etc., indicating their key roles in regulating immune responses. In the S3D Fig, we found that there was significant communication between monocytes and regulatory T cells, M1 macrophages, and CD8+ T cells, suggesting an important role in the immune system. S3E Fig showed the communication network of M2 macrophages. There was a strong communication link between M2 macrophages and regulatory T cells, fibroblasts and NK cells, indicating their function in tissue repair and anti-inflammatory response. As revealed in the S3F–S3K Fig, there was the communication networks of M1 macrophages, fibroblasts, NK cells, B cells, CD8+ T cells, and CD4+ memory T cells, respectively, revealing the important role of these cell types in inflammatory response, tissue repair, antibody production, cytotoxic response, and long-term immune memory.

Through the above analysis, we mapped the communication networks of multiple cell types in the immune microenvironment, revealing the complex interactions between different cell types. These results provide a new perspective for understanding the regulatory mechanisms of immune response and provide a theoretical basis for potential immunotherapeutic targets.

We further analyzed intercellular communication networks using single-cell RNA sequencing data, with a particular focus on the roles of ligand-receptor pairs in mediating cell-to-cell interactions. S4 Fig illustrates the probability and significance of communication between various ligand-receptor pairs across different cell types. The intensity of communication is represented by color, with darker shades indicating higher probabilities. Notably, ligand-receptor pairs such as ANGPTL2-ITGA5+ITGB1, ANGPTL2-PIRB, and ANGPTL2-TLR4 exhibited high communication probabilities between regulatory T cells and other cell types. Additionally, ligands including ANGPTL4, ANXA1, AREG, BDNF, BMP2, and BMP4 demonstrated significant interactions with their respective receptors.

Statistical analysis confirmed the significance of several ligand-receptor pairs in intercellular communication, particularly among immune cells, including CCL3-CCR1, CCL5-CCR5, CXCL12-CXCR4, and HBEGF-EGFR. These findings underscore the pivotal roles of specific ligand-receptor pairs in regulating immune cell interactions and signaling pathways. This comprehensive analysis enhances our understanding of the complex communication networks within the cellular microenvironment and offers potential targets and theoretical foundations for future immunotherapies and disease interventions.

### The communication of MIF signal networks between different cell types

We further analyzed the Macrophage Movement Inhibitor (MIF) signaling network between different cell types. It was revealed that regulatory T cells and M2 macrophages were important senders and receivers of MIF signaling network, suggesting their important role in the immune regulation (Fig 3A). The intercellular communication network between MIF and the ligand-receptor pairs of CD74-CXCR4 and CD74-CD44 was shown, respectively (Fig 3B and 3C). Regulatory T cells, B cells, and CD8+ T cells showed significant communication links in these two communication networks, indicating the critical role of these cell types in MIF signaling. The communication network of the ligand-receptor pair of IF-ACKR3 further confirmed the importance of regulatory T cells and other immune cells in the MIF signaling network (Fig 3D). Regulatory T cells show high importance in various roles, especially in sending and receiving signals, which further supports their central role in immune regulation (Fig 3E). These results suggest that the MIF signaling network plays an important role in regulating communication between immune cells. Regulatory T cells, M2 macrophages, and other immune cells interact extensively through the MIF signaling network, providing new insights into the complex regulatory mechanisms of immune responses and providing potential targets and theoretical foundations for future immunotherapy research [16,19,20].

As revealed in the S5A Fig, regulatory T cells, M2 macrophages and B cells have significant communication links in this network, indicating that these cell types play an important role in MIF-ACKR3 signal transduction. In the communication

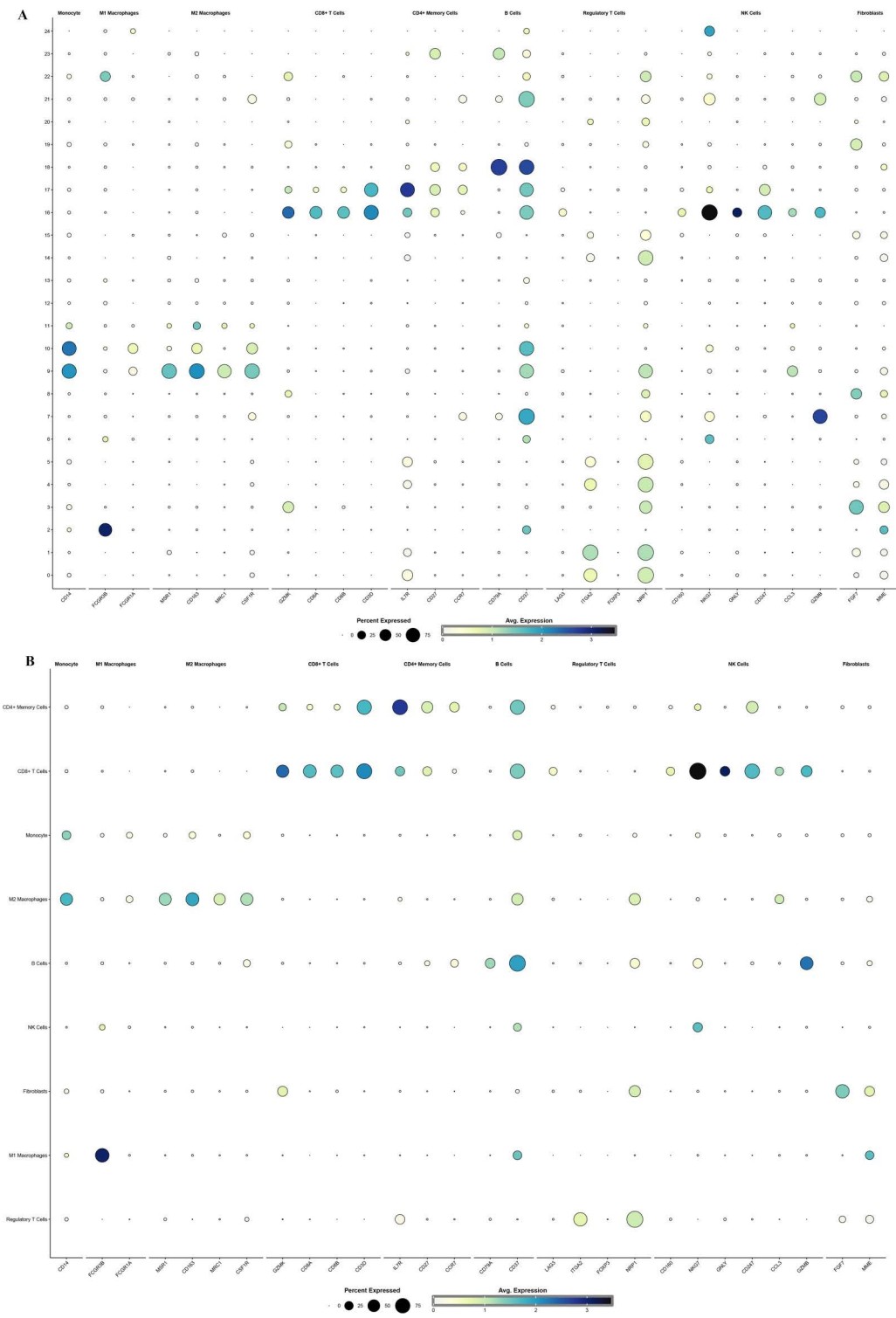

**Fig 3. The MIF signal path network analysis.** (A) The heatmap of the MIF signal network. (B) The receptor and ligand interaction of CD74 and CXCR4 in the MIF signal network was shown. (C) The receptor and ligand interaction of CD74 and CD44 in the MIF signal network was shown. (D) The receptor of ACKR3 in the MIF signal network. (E) Network centrality scores of MIF are shown.

network between MIF and ligand-receptor pair of CD74+CD44, regulatory T cells, CD4+ memory T cells and B cells showed significant communication links in this network, indicating that these cells play a key role in MIF-CD74+CD44 signal transduction (S5B Fig). As for ligand-receptor pair of CD74+CXCR4, regulatory T cells, CD8+ T cells and CD4+ memory T cells have strong communication links in this network, which further confirms the importance of these cells in MIF-CD74+CXCR4 signal transduction (S5C Fig). These results indicate that MIF communicates with various immune cell types through different receptors, especially in regulatory T cells, M2 macrophages and B cells. MIF signaling network plays an important role in the complex interaction between immune cells, providing a new perspective for understanding the regulation mechanism of immune response, and providing potential targets and theoretical basis for future immunotherapy research.

### Identification of DEGs between OP and normal bone tissues in bulk-RNA data

Based on the DEGs analysis of bulk-RNA data from GEO data (GSE56116 and GSE62402), we identified the genes with significant changes between OP and normal controls. The volcano plot showed the expression changes of these DEGs, in which the red dots represent genes that are significantly upregulated, the blue dots represent genes that are significantly downregulated, and the gray dots represent genes that have no significant changes (Fig 4A). The expression matrix of significantly different genes in different samples were showed by the heatmap (Fig 4B). Fig 4C shows the Venn diagram of DEGs and oxidative stress (OS) related genes. The results showed that 53 genes appeared in the DEGs and OS gene set at the same time, suggesting that these genes may have a common regulatory mechanism in specific biological processes.

Through the enrichment analysis of GO and KEGG pathway of DEGs, we further found the functions of these DEGs in the biological process. The GO enrichment analysis of shows that these genes are significantly enriched in many biological processes and molecular functions, such as the positive regulation of cytokine production, the response to bacteria-derived molecules, and the positive regulation of inflammatory response (Fig 4D). The results of KEGG pathway enrichment analysis revealed that these genes are significantly enriched in many biological pathways, including lipid and atherosclerosis, neutrophil extracellular network formation, and NOD-like receptor signaling pathway (Fig 4E). These results show that DEGs play an important role in many biological processes and signal pathways, which provides important clues for us to understand the role of these genes in disease mechanism and biological function.

### Immune microenvironment analysis and identification of vital genes

The relative proportions of immune cell types between OP and normal bone tissues were analyzed, revealing significant differences in immune cell composition between the disease and control groups (Fig 5A). Notably, variations in T cells, B cells, and macrophages suggest a substantial impact of disease state on immune cell populations.

LASSO regression analysis was performed to identify optimal model parameters based on the variation in cross-validation mean square error (RMSE), highlighting the significance of DEGs in predicting disease states (Fig 5B, 5C). Among these, CHRM2 demonstrated strong predictive ability with an area under the curve (AUC) value of 0.750 in the ROC analysis, underscoring its potential as a biomarker for distinguishing disease from healthy states (Fig 5D).

KEGG pathway enrichment analysis, conducted through gene set variation analysis (GSVA), revealed significant enrichment of multiple signaling pathways under varying experimental conditions, including tryptophan metabolism, drug metabolism, amino acid metabolism, and the complement and coagulation cascade (Fig 5E). These pathways are likely critical in the disease state and therapeutic response.

Collectively, the findings demonstrate that treatment significantly influences immune cell composition and gene expression profiles. Through LASSO regression and ROC analysis, key genes such as CHRM2 were identified as potential predictors of OP disease status. Furthermore, GSVA analysis provided insight into the involvement of these DEGs in

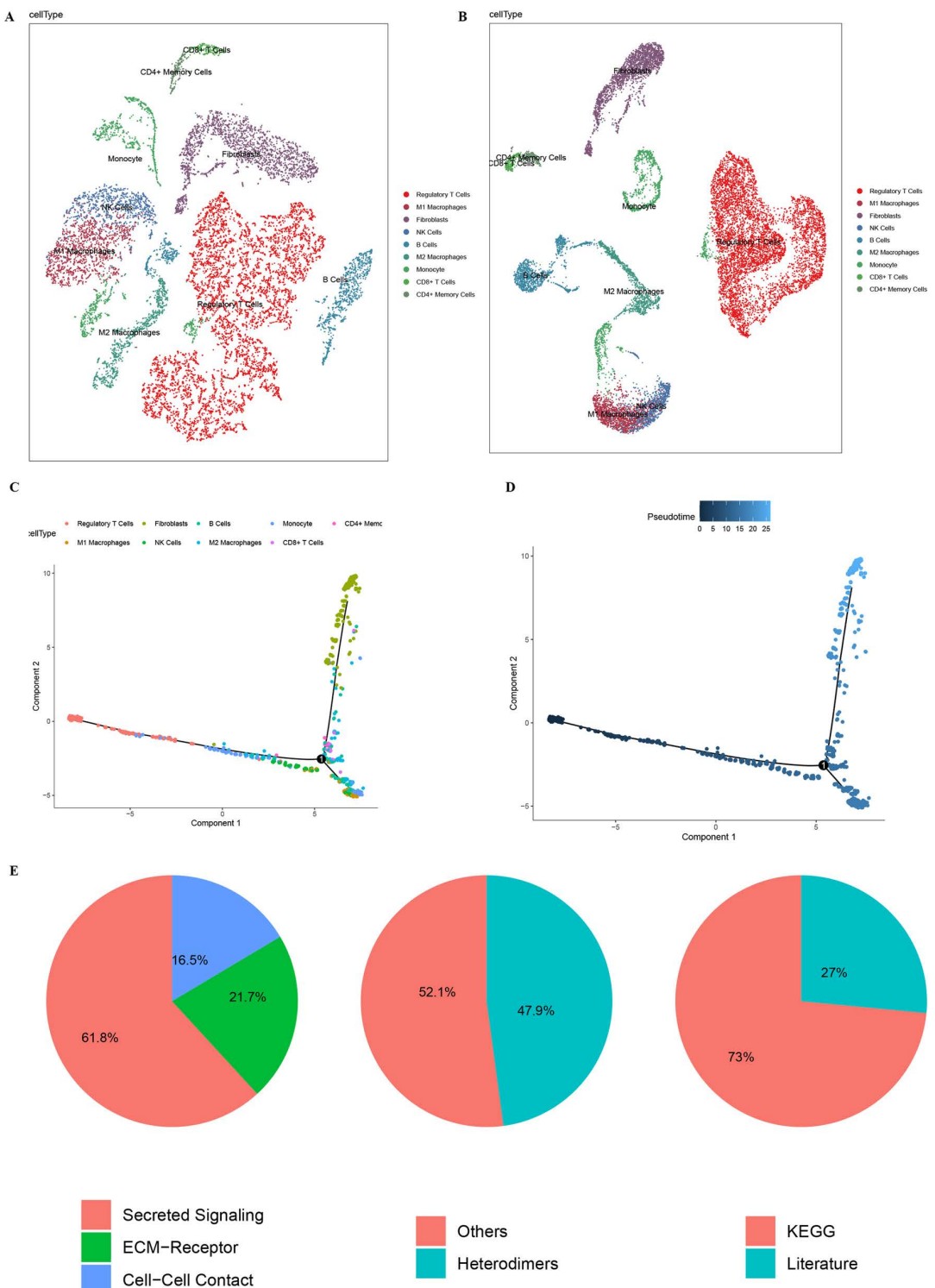

**Fig 4. The identification of DEGs and functional enrichment analysis.** (A) The volcano Plot showed DEGs between the OP samples and normal bone tissues. (B) The heatmap showed the expression levels of DEGs in different sample groups. (C) The Venn Diagram showed the overlap genes of oxidative stress-related genes (OS) and DEGs. GO (D) and KEGG (E) pathway functional enrichment analysis of DEGs.

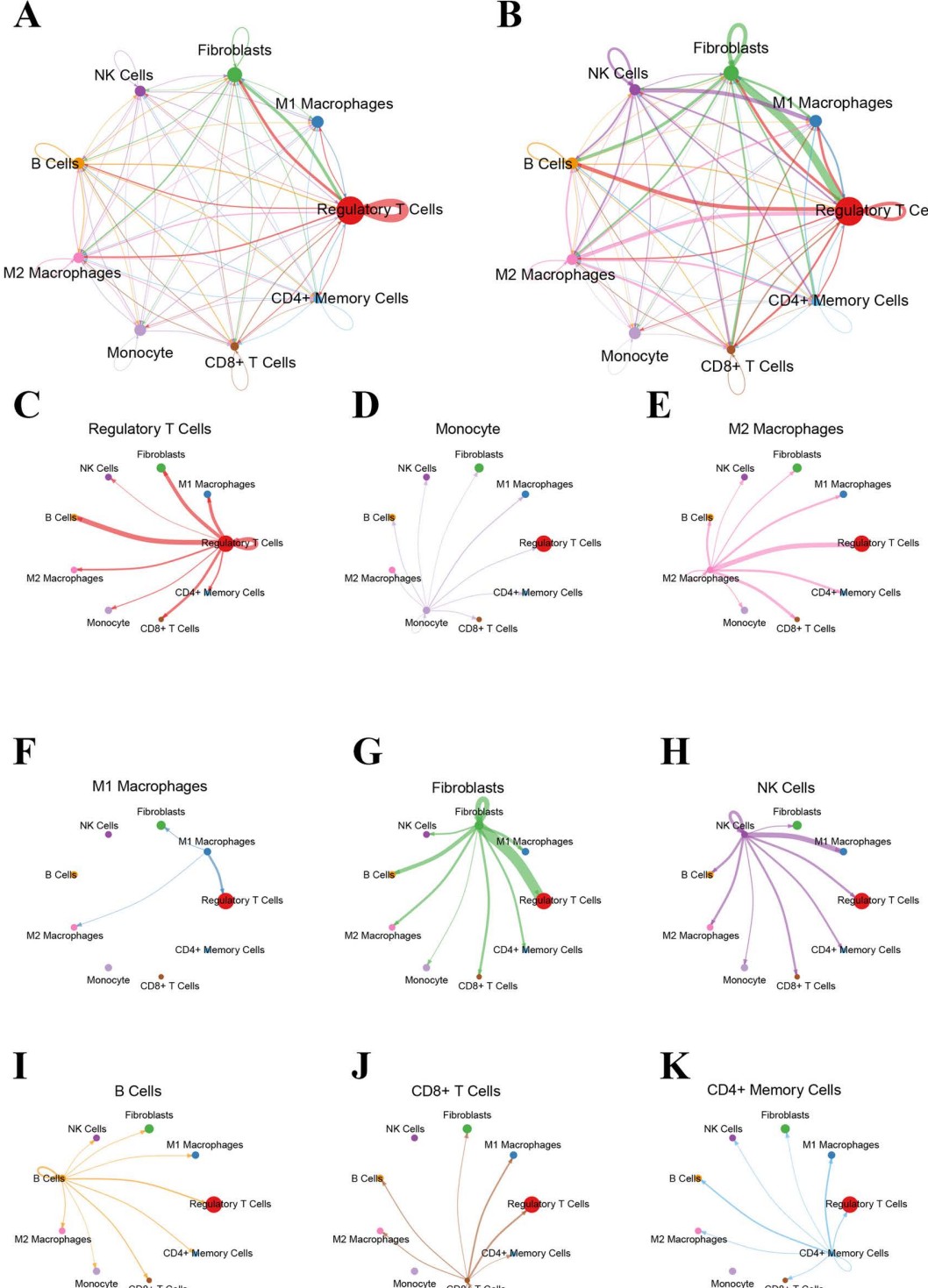

**Fig 5. Cell components and correlation analysis.** (A) The relative percentage of different immune cell types in the treat and control group. (B) The LASSO coefficient distribution. (C) The cross-validation curves. (D) The ROC curve of CHRM2 to distinguish the osteoporosis from the control group. (E) The GSVA functional enrichment analysis.

essential biological pathways, offering valuable clues for understanding the molecular basis of OP and its therapeutic mechanisms.

## Expression level of CHRM2 in OGD process

We analyzed the expression levels of CHRM2 in the osteogenic differentiation (OGD) model of HUM-iCELL-s011 cells. The ALP staining and alizarin S staining were performed at different time points of osteogenic differentiation (day 0, 7, 14, and 21) to evaluate the changes in mineralized nodule content and ALP activity. The results showed that with the progression of differentiation, the content of mineralized nodules and ALP activity gradually increased, reflecting the progression of osteogenic differentiation (Fig 6A, 6B). The Western blot and qPCR results showed that OGD-related factors, such as RUNX2 and OPN [21–24] were highly expressed during the OGD, which was consistent with key regulatory mechanisms during osteogenic differentiation. The expression of CHRM2 showed a significant time-dependent change: mRNA and protein levels of CHRM2 were significantly higher on day 7 than on day 0, indicating up-regulation in the early differentiation stage. However, at day 14, mRNA and protein levels of CHRM2 did not show significant differences compared with day 7, suggesting that its expression tended to stabilize in the medium term. By day 21, mRNA and protein levels of CHRM2 were down-regulated, which may be related to regulatory mechanisms in the late differentiation stage (Fig 6C, 6D). In summary, the dynamic expression of CHRM2 in the OGD model is closely related to the process of osteogenic differentiation, which provides important clues for further exploration of its specific function in osteogenic differentiation.

## Inhibition of CHRM2 can inhibit cell proliferation and cell cycle, but promote OGD process

We transfected siRNA into HUM-iCELL-s01 cells to knockdown CHRM2, which was verified by Western blot and qPCR results (Figs 7A, 7B). The CCK-8 results showed that the knockdown of CHRM2 significantly inhibited cell proliferation, suggesting that CHRM2 may play a key role in the regulation of cell proliferation (Fig 7C). To further verify this result, we analyzed the vital proteins related to osteogenic differentiation by Western blot analysis. The results showed that the protein expression levels of RUNX2, OPN, and Osterix were also significantly higher in CHRM2 knocked down cells than in the control group, further supporting the role of CHRM2 as a negative regulator during osteogenic differentiation (Fig 7D). In order to further explore its impact on the cell cycle, flow cytometry analysis was performed and revealed that the proportion of cells in the G1 phase of the Si-CHRM2 experimental group was significantly increased compared with the negative control group, while the proportion of cells in the S phase was significantly decreased, suggesting that CHRM2 knockdown may lead to cell cycle stagnation in the G1 phase (Fig 7E) and inhibited the cells from entering the DNA synthesis phase. To evaluate the effect of CHRM2 knockdown on osteogenic differentiation, ALP activity and alizarin red staining showed that Si-CHRM2 significantly increased ALP activity and mineralized nodule formation compared with negative controls. These results suggests that down-regulation of CHRM2 may promote the osteogenic differentiation potential of cells. Our study suggests that CHRM2 plays an important role in regulating the proliferation and osteogenic differentiation of HUM-iCELL-s01 cells. Its knockdown not only inhibited cell proliferation, but also enhanced osteogenic differentiation by promoting the expression of osteogenic genes and proteins.

## Gene expression profile of mRNA binding CHRM2

In order to elucidate the molecular mechanism of CHRM2-mediated regulation in HUM-iCELL-s01 cells, we performed RNA Immunoprecipitation (RIP) assay and sequencing in the CHRM2 knockdown cells. The FPKM were used for the normalization of the raw data (Fig 8A and 8B). With IP/input >2 and p < 0.05 as the standard, a total of 216 DEGs were identified by RIP sequencing analysis. DEGs were enriched in graphene oxide and analyzed according to their

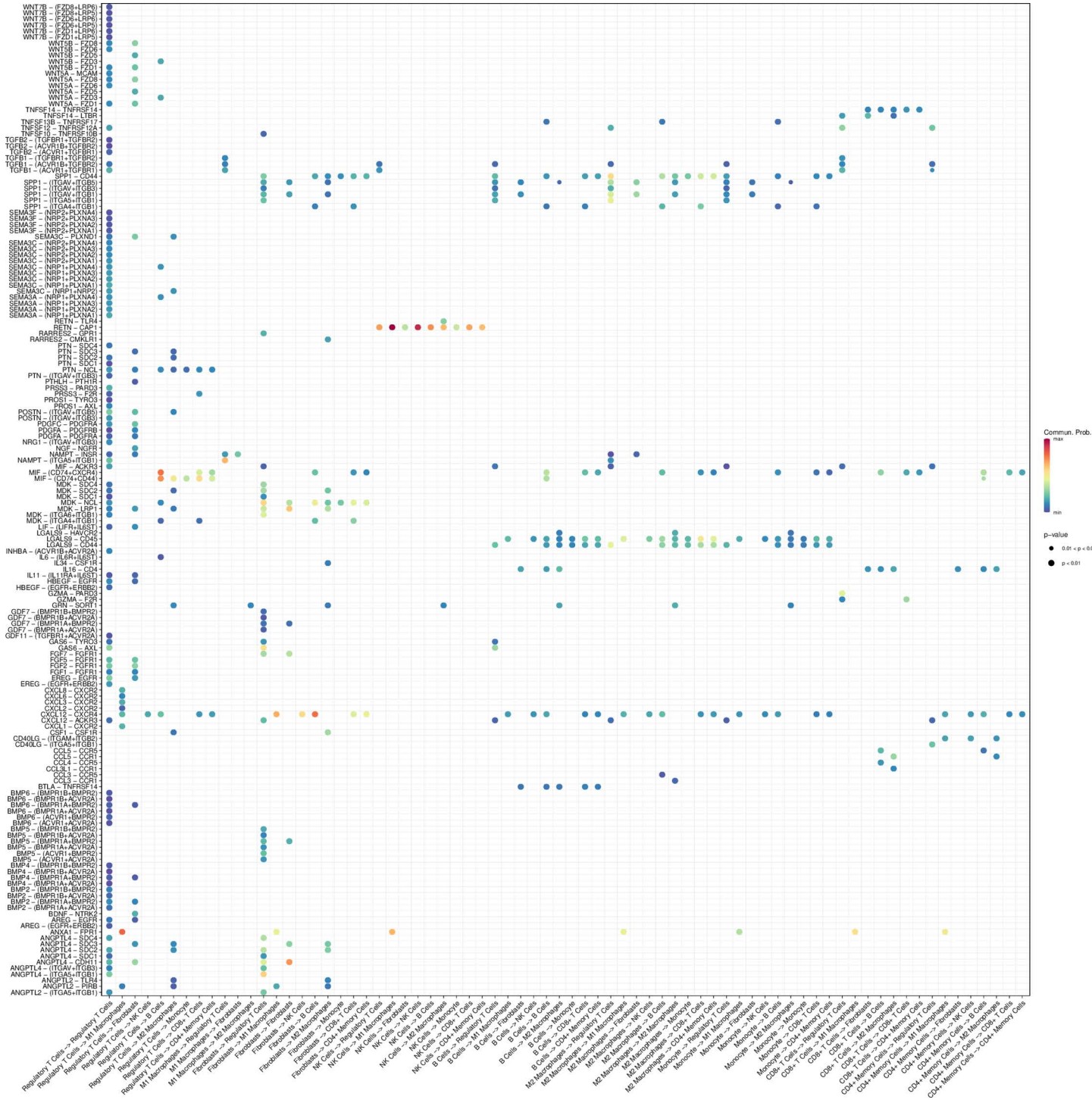

**Fig 6. The expression levels of CHRM2 and OGD-related factors at 0, 7, 14 and 21 days after OGD induction.** The ALP staining (A), alizarin red staining (B), western blot (C) showed protein levels of CHRM2 and OGD-related factors (OPN and RUNX2) at 0, 7, 14, and 21 days after OGD induction, normalized with GAPDH. (D) The mRNA expression levels of OGD-related factors (OPN and RUNX2) was detected by RT-qPCR and normalized with GAPDH. The results were shown as mean ±SD. *p < 0.05; **p < 0.01; ***p < 0.001; ns: no significance.

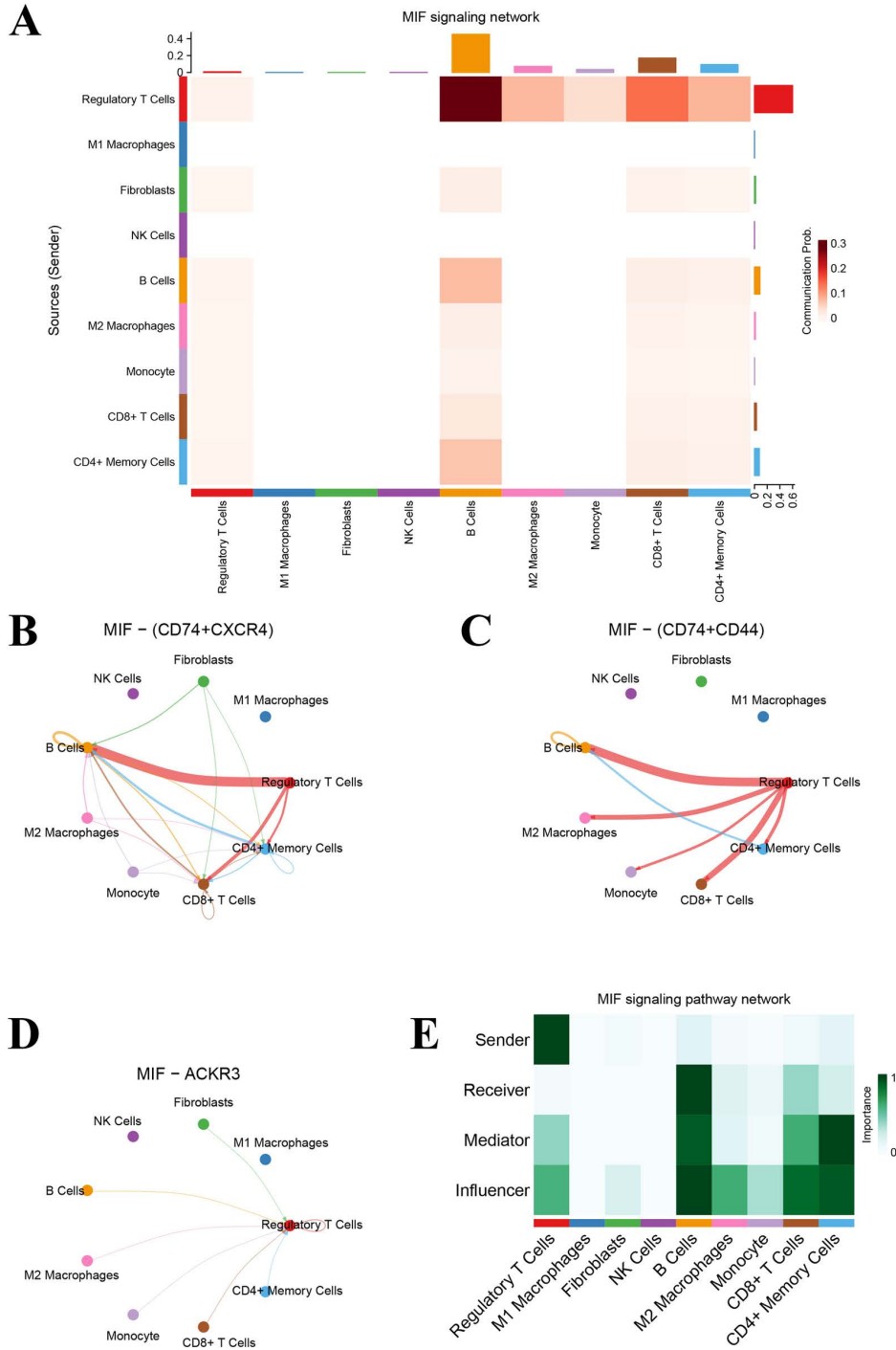

**Fig 7. The effects of CHRM2 knockdown on osteogenic differentiation and cell proliferation.** (A) The expression of CHRM2 was significantly reduced in cells knocked down with Si-CHRM2. (B) The RT-qPCR results showed that the expression of CHRM2 in cells with Si-CHRM2 knockdown decreased significantly. (C) The cell proliferation were measured by OD value at 450 nm, and the observation time points were at 0, 24, 48, 72 and 96 hours. (D) The protein expression levels of osteogenic markers RUNX2, OPN, and Osterix in Si-CHRM2 knockdown cells and control cells by western blot. (E) Flow cytometry of cell cycle in the control group and the experimental group. (F) ALP staining in the Si-CHRM2 experimental group. (G) Alizarin red staining. *p < 0.05; **p < 0.01; ***p < 0.001; ns: no significance.

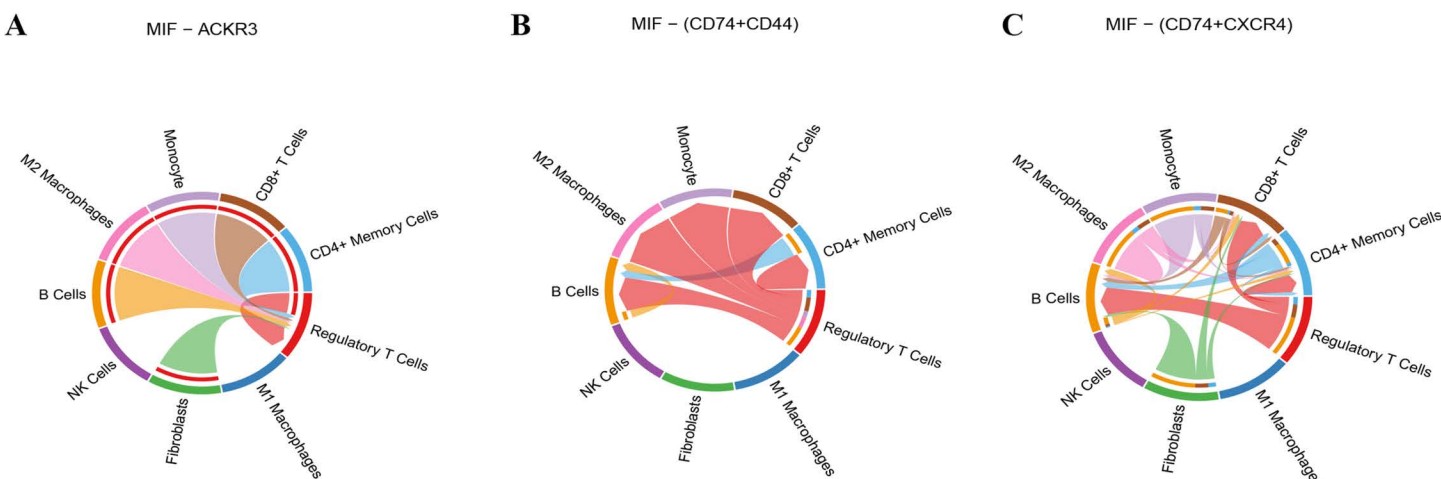

**Fig 8. CHRM2 binding mRNA expression profile.** (A) Box plot of FPKM values showing gene expression levels between BMSC1-IP and BMSC1-input groups. (B) Violin plots of FPKM values further demonstrated the distribution of gene expression between BMSC1-IP and BMSC1-input groups. (C) The heat map shows the correlation between BMSC1-IP and BMSC1-input samples. *p < 0.05; **p < 0.01; ***p < 0.001.

involvement in Biological processes (BP), Molecular functions (MF) and Cellular components. CC) and other categories for classification and annotation. In the CC category, the terms identified include cellular parts, cells, intracellular parts, organelles, and endoplasmic binding organelles. For BP, processes such as cell regulation, cell metabolism regulation, primary metabolism and metabolic processes of organic matter were identified. For MF, terms such as binding and protein binding are identified. KEGG analysis was used to identify the major biochemical metabolic and signal transduction pathways associated with these protein-binding genes. Relevant signaling pathways include thyroid hormone, mitogen-activated protein kinase (MAPK), neurotrophin factor, mechanism target of rapamycin (mTOR), Forks Box O (FoxO), sphingolipid, phosphocreatine 3 kinase (PI3K) -Akt, stem cell pluripotency, and Wnt Chronic depression, lysine degradation, adhesion, and melanin production. The results of GO analysis suggested that DEGs were enriched in BP in receptor aggregation, excitatory state response, endocytosis and cytoskeletal remodeling. These processes were closely related to cell signal transduction, inflammatory response and cell morphological changes (Fig 9A). KEGG pathway analysis results indicated that DEGs were significantly enriched in several pathways, such as extracellular matrix-receptor interaction, basement membrane, and extracellular matrix tissue remodeling, which were closely related to the dynamic changes of extracellular microenvironment and the process of tumor invasion and metastasis (Fig 9B). The results of GO analysis suggested that DEGs were enriched in the functional categories of actin binding, calcium ion binding, and cell adhesion molecule binding in MF, suggesting that these genes play an important role in cytoskeleton regulation, calcium signaling and cell interactions (Fig 9C). The results of KEGG pathway analysis indicated the enrichment of multiple signaling pathways of DEGs, including ECM-receptor interaction, local adhesion, protein folding and transport, suggesting that these DEGs may play an important role in tumor cell adhesion, migration, and protein homeostasis maintenance (Fig 9D). CDC80, COL4A2, DBN1, FLNA, HSPG2, MPRIP, MYH9, MYH10, PLEC, and TNRC18 were included in DEG, and the $\log_2$FC value of COL4A2 was the highest among all DEGs. The RIP-qPCR showed the binding of CHRM2 to the above RNAs (Fig 9E).

## Effect of CHRM2 on COL4A2 expression level

To further investigate the relationship between the CHRM2 and COL4A2, we performed qPCR and western blot and found that CHRM2 down-regulation could down-regulate COL4A2 expression level (Fig 10A, 10B). Then, we detected

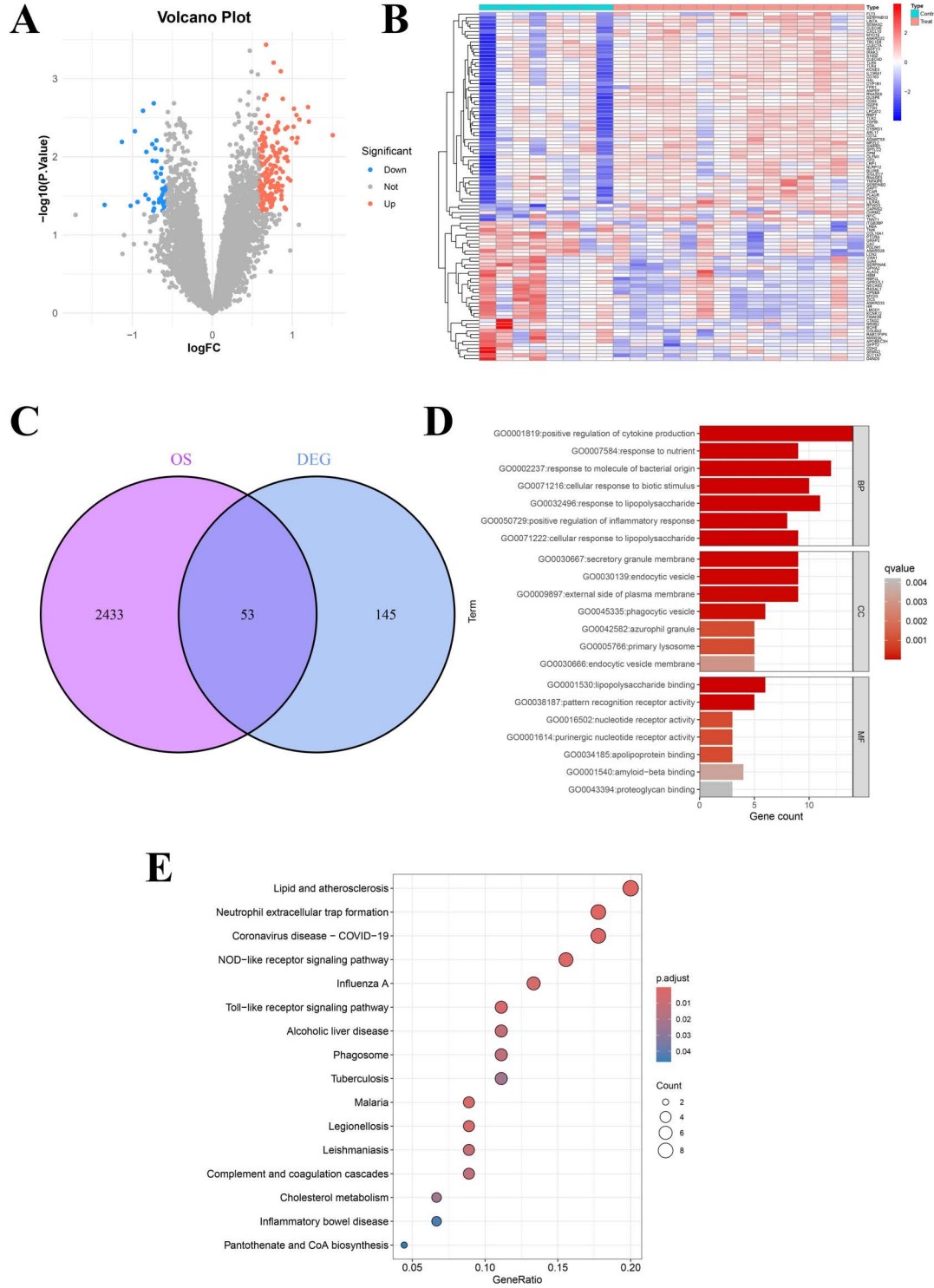

**Fig 9. GO functional annotation and KEGG pathway enrichment analysis of DEGs.** BP (A), CC (B), and MF (C) enrichment analysis of GO in DEGs. (D) KEGG pathway enrichment analysis results of DEGs. (E) Thesignificant DEGs between the IgG and IP group were shown, which indicated the highest log2FC of COL4A2. *p < 0.05; **p < 0.01; ***p < 0.001.

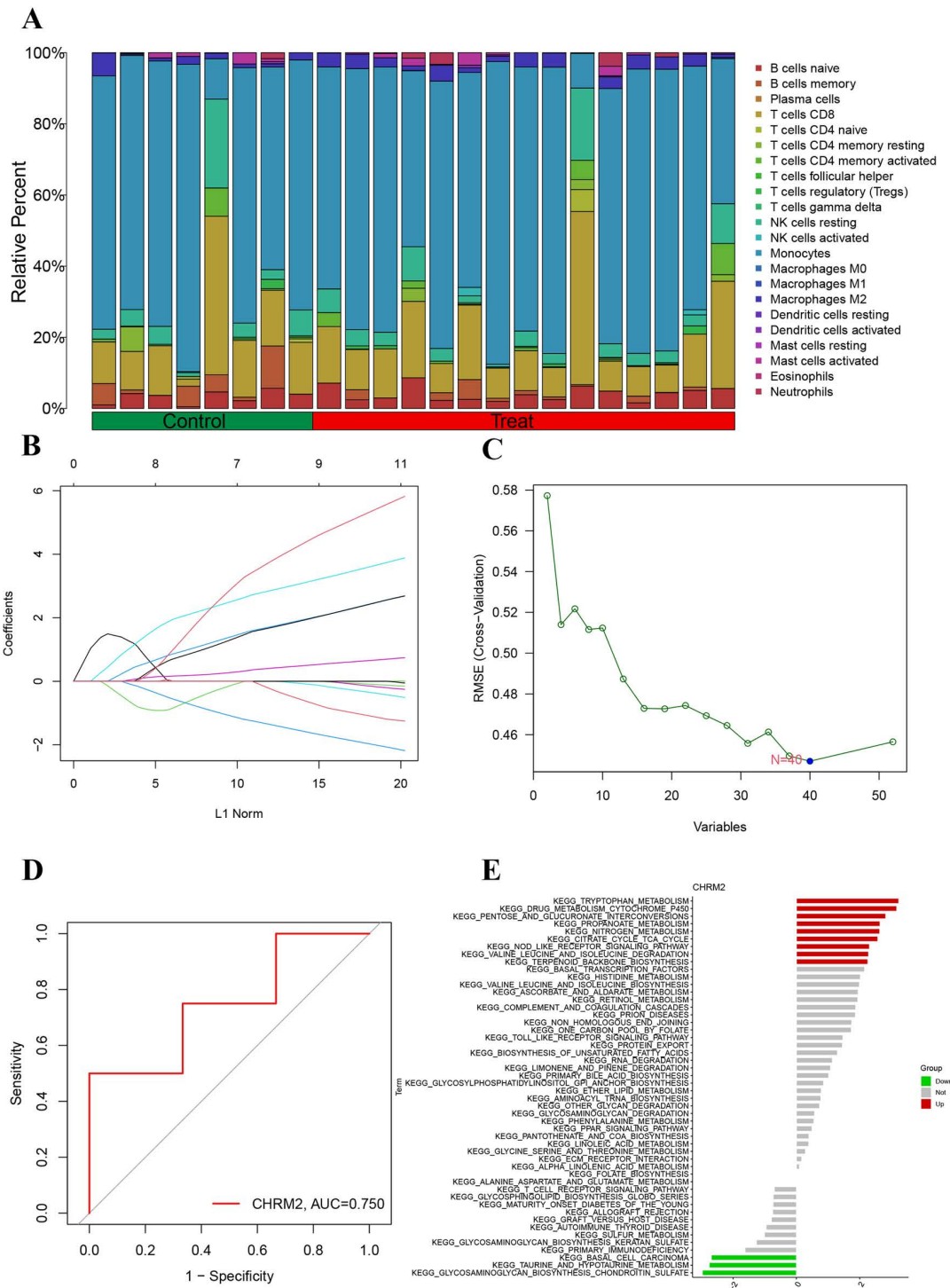

**Fig 10. Effect of CHRM2 knockdown on the COL4A2 expression level.** (A) The protein expression level of COL4A2 in the Si-CHRM2 and control cells. (B) The results of RT-qPCR showed the expression of COL4A2 mRNA in si-CHRM2 knockout at different time points (0 h, 6 h, 12 h). (C) Immunofluorescence staining images showed COL4A2 (red) staining results at different time points (0 h, 6 h, 12 h) in the control and Si-CHRM2 groups. *p < 0.05; **p < 0.01; ***p < 0.001.

the co-localization of CHRM2-COL4A2 by IF and FISH. The mRNAs of both were expressed in the nucleus and cyto-plasm, while the COL4A2 protein was localized in the nucleus. IF results confirmed that COL4A2 protein levels decreased with CHRM2 levels, and FISH showed that COL4A2 mRNA levels decreased with CHRM2 levels (Fig 10C).

## COL4A2 regulates cell proliferation, cell cycle and OGD process

We further evaluated the biological function of COL4A2 in HUM-iCELL-s01 cells, and analyzed the expression of vital proteins associated with osteogenic differentiation by Western blot. The results showed that the protein expressions of RUNX2, OPN and Osterix in CHRM2 knocked down cells were also significantly higher than those in the negative control group, further supporting the role of CHRM2 as a negative regulator during osteogenic differentiation (Fig 7D). It was also revealed that the expression of COL4A2 increased on the 7th day. However, it decreased on day 14 (the baseline level was day 0), and the expression of COL4A2 protein on day 14 was equal to that on day 0, consistent with the change in CHRM2 expression in OGD (Figs 11A, 11B). Further experiments verified the mutual regulation of COL4A2 and CHRM2 in cells. In the Si-NC and Si-COL4A2 group, we observed that the expression of CHRM2 was not changed when COL4A2 was knocked down (Fig 11C, 11D). Immunofluorescence showed that the fluorescence intensity of COL4A2 and CHRM2 was significantly weakened in the si-COL4A2 group (Fig 11E, 11F), further supporting their co-expression relationship and interaction. Quantitative analysis also showed that the fluorescence intensity of CHRM2 and COL4A2 decreased significantly after CHRM2 was knocked down. These results indicated that the expression level of COL4A2 might be regulated by CHRM2.

The expression of key proteins involved in osteogenic differentiation was further analyzed using qPCR and Western blot. The results indicated that the levels of RUNX2, OPN, and Osterix were significantly higher in cells with COL4A2 knockdown compared to the control group, highlighting the role of COL4A2 as a negative regulator of osteogenic differentiation (Fig 12A). Similarly, mRNA expression of OCN was markedly upregulated in the si-COL4A2 group (Fig 12B). These findings suggest that COL4A2 knockdown not only inhibits cell proliferation but also enhances osteogenic differentiation by upregulating critical osteogenesis-related proteins.

CCK-8 assays revealed that COL4A2 knockdown significantly reduced cell proliferation, indicating its potential role in regulating the cell proliferation process (Fig 11C). Flow cytometry analysis showed that cells with COL4A2 knockdown were predominantly arrested in the G1 phase, with a significantly lower proportion of cells in the S and G2 phases, suggesting that COL4A2 knockdown disrupts normal cell cycle progression from G1 to S phase (Fig 11D).

Additionally, COL4A2 knockdown induced osteogenic differentiation in HUM-iCELL-s01 cells. ALP staining and activity assays demonstrated significantly increased ALP activity in COL4A2 knockdown cells compared to controls, indicating that COL4A2 knockdown promotes osteogenic differentiation (Fig 11E). Alizarin red staining and quantitative analysis further confirmed that mineralized nodule formation was significantly enhanced in the si-COL4A2 group, reinforcing the negative regulatory role of COL4A2 in osteogenic differentiation (Fig 11F).

In summary, this study reveals a critical regulatory role for COL4A2 in cell proliferation and osteogenic differentiation in HUM-iCELL-s01 cells. The knockdown of COL4A2 not only impairs normal cell cycle progression but also promotes the expression of osteogenic genes and the formation of mineralized nodules. These findings provide a valuable theoretical foundation for further exploration of COL4A2 in bone biology.

## Discussion

Osteoporosis (OP) is a systemic bone disease caused by dyshomeostasis, characterized by decreased bone mass and degradation of the microstructure of bone tissue, usually resulting in fracture in the absence of trauma or minor trauma, or causing pain, deformity, dysfunction, and possibly death [25]. With the acceleration of the global aging process, the prevalence of OP is rising, which has become one of the main problems, threatening the health and quality of life of the

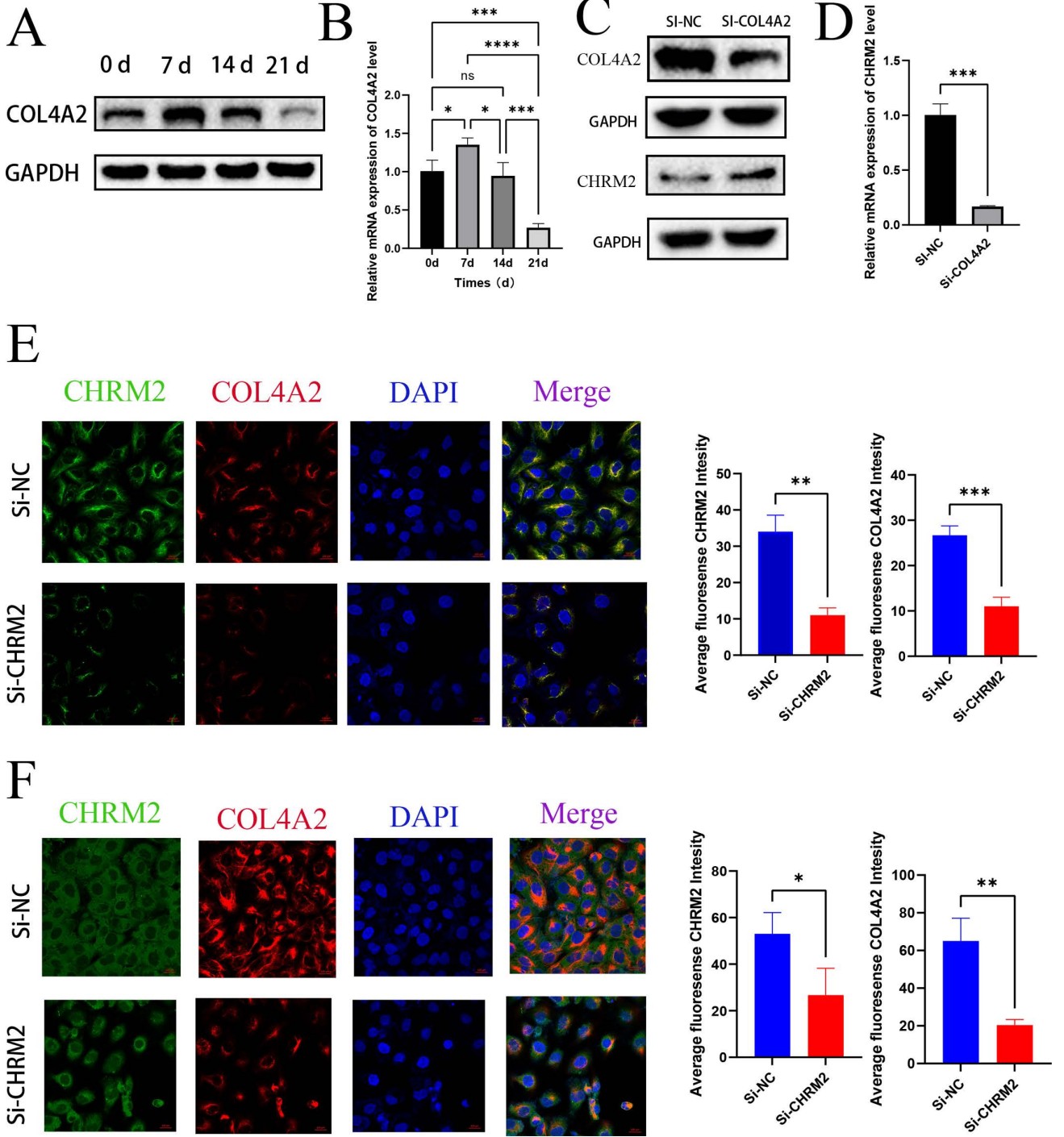

**Fig 11. The expression level of COL4A2 at different periods and the effect of COL4A2 knockdown on cells.** (A) The protein expression level of COL4A in the OGD model at 0, 7, 14, and 21 days. (B) The mRNA expression levels of COL4A2 at different time points of OGD (0, 7, 14, and 21 days); (C) The protein expression levels of COL4A2 and CHRM2 in the cells of the control group and the Si-COL4A2 group. (D) The relative mRNA expression level of CHRM2 in the control and si-COL4A2 experimental groups. (E) Immunofluorescence staining images showed the expression and localization of CHRM2 (green) and COL4A2 (red) proteins in cells of the control and si-COL4A2 groups, while DAPI (blue) was used for nuclear staining. (F) FISH and IF images further showed the expression changes of CHRM2 mRNA and COL4A2 mRNA. *p < 0.05; **p < 0.01; ***p < 0.001.

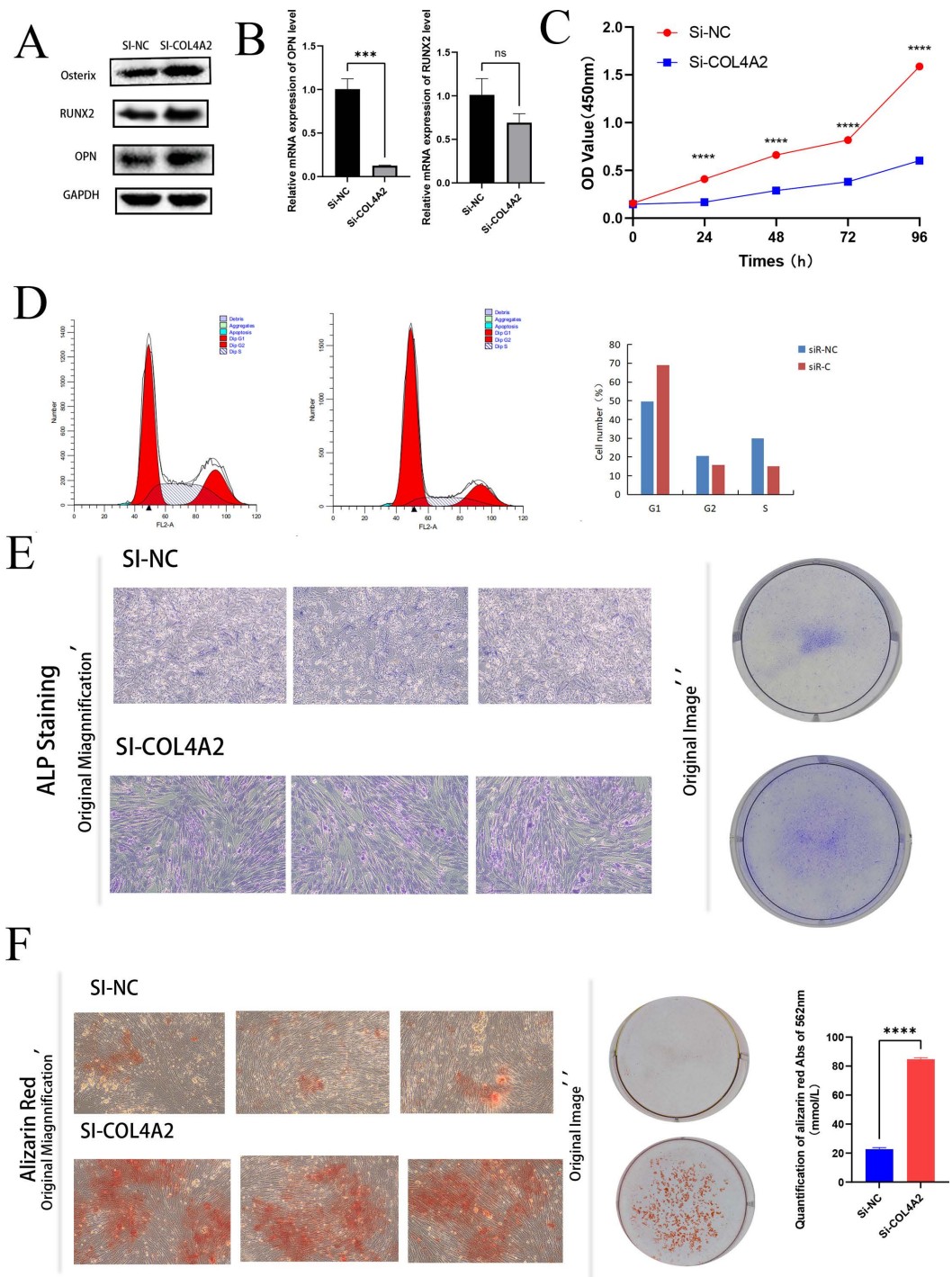

**Fig 12. Effect of COL4A2 knockdown on osteogenic differentiation and cell proliferation.** (A) The expression levels of Osterix, RUNX2, and OPN in the control and si-COL4A2 groups by western blot. (B) The mRNA expression levels of OPN and RUNX2 in the control group and the si-COL4A2 groups. (C) Cell proliferation assay (CCK-8) showed OD values at 450 nm at different time points (0, 24, 48, 72 and 96 hours) in the control group and the si-COL4A2 group. (D) The cell cycle distribution in the control and si-COL4A2 groups was analyzed by flow cytometry. (E) ALP staining in the control and si-COL4A2 group. (F) The alizarin red staining showed mineralized nodules formation in the control and si-COL4A2 groups. *p < 0.05; **p < 0.01; ***p < 0.001.

elderly. The pathogenesis of this disease is complex, involving many factors such as genetics, hormones, nutrition and lifestyle [26]. The current treatment strategies mainly include drug therapy and lifestyle intervention, but the efficacy of the drug is limited, accompanied by certain side effects [27]. Therefore, exploring the pathological mechanism of OP and its key regulatory factors is of great significance for early diagnosis and treatment.

During the last few years, the in-depth mining of databases, such as the GEO database containing clinicopathological information and gene expression levels of patients, has been flourishing and aided researchers in discovering biomarkers and therapeutic targets [28]. Therefore, we dig into the information of GEO data to discover the key predictor gene for OP patients. We analyzed scRNA data of OP samples and normal bone tissues from the GEO database [29], and labeled different cell types based on the known markers, including monocytes, M1 macrophages, M2 macrophages, CD8+ T cells, CD4+ memory T cells, B cells, regulatory T cells, NK cells, and fibroblasts. There were significant differences in the distribution of different cell types in OP patients and normal bone tissue. We found that the proportion of M1 macrophages and regulatory T cells increased significantly in OP patients, while the proportion of CD4+ memory T cells and fibroblasts was higher in normal bone tissue. These differences may reflect the pathological changes and immune responses in osteoporosis. By using t-SNE and UMAP dimensionality reduction analysis, we further confirmed the significant separation of different cell types in spatial distribution. These results showed that there were significant differences at the gene expression level between normal bone tissue and cells of patients with osteoporosis, providing important clues for our understanding of the pathogenesis of osteoporosis.

The cell trajectory analysis revealed that osteoporosis-associated cell types, such as M1 macrophages and regulatory T cells, showed specific evolutionary paths along the pseudo-timeline that may be closely related to the progression of the disease. Functional enrichment analysis showed that genes related to secretion signaling, extracellular matrix receptor interactions, and cell-cell contact were significantly altered in the cells of patients with OP, and changes in these genes may play a key role in the development of OP. In the construction and analysis of cell communication network, we found that regulatory T cells have significant centrality in the network, indicating their important role in intercellular communication. Regulatory T cells have strong communication with a variety of cell types, including M2 macrophages, monocytes and fibroblasts, suggesting that they play a key role in the regulation of immune responses. In addition, the analysis of MIF signaling networks further revealed the importance of regulatory T cells and M2 macrophages in immune regulation [30]. Our scRNA-seq data corroborate earlier findings that the osteoporotic bone microenvironment features an increase in pro-inflammatory cells, notably M1 macrophages. However, our study extends these observations by uncovering a significant rise in regulatory T cells in OP lesions.

In 2000, Arron and Choi et al. [31] proposed skeletal immunology, which studied the interaction between the immune system and bone, and became a new direction in the pathogenesis of OP. More and more clinical studies have confirmed that chronic inflammation was closely related to OP, which may be related to age-related oxidative stress and low immune system activation [32]. With age, the body is in a state of continuous oxidative stress and low activation of the immune system, resulting in dysfunction of the T and B lymphocyte system and disruption of the balance between inflammatory factors and protective immune factors [33]. T cell subsets and their activation pathways were significantly altered, resulting in enhanced bone resorption and reduced bone formation. T lymphocytes can be divided into CD4+T cells, CD8+T cells and regulatory T cells. CD4+T cells, also known as T helper cells, secrete a variety of cytokines that interact with other immune cells and bone cells. In postmenopausal patients, the level of Th cells increased significantly and secreted a large number of cytokines such as IL-1 and TNF-alpha, which participated in the conduction of RANK signaling pathway and regulated the maturation and differentiation of osteoclasts, thus promoting the occurrence and progression of postmenopausal OP [34]. The effect of oxidative stress on osteoporosis is particularly important, as it may be one of the underlying causes of osteoblast and osteoclast dysfunction [35]. The osteogenic differentiation of mesenchymal stem cells is crucial for maintaining the balance of bone metabolism [36]. Oxidative stress interferes with this process, which may lead to the reduction of new bone formation, destroy the balance between bone formation and bone resorption, and further aggravate

osteoporosis. Therefore, in order to further study the pathogenesis of OP, we used Bulk-RNA transcriptome data for further analysis, and screened 198 DEGs between OP samples and normal bone tissues. Through the intersection screening of genes related to oxidative stress, 53 vital genes closely related to OP and oxidative-stress were finally identified. Then, we used LASSO regression analysis and machine learning methods to identify CHRM2 as the core gene [37]. CHRM2 shows significant potential in predicting disease states of OP. We also found that the AUC value of CHRM2 gene reached a high level, showing its excellent performance in predicting OP.

Based on GSVA results, we found that tryptophan metabolism, drug metabolism, amino acid metabolism, complement and coagulation cascade play an important role in the occurrence and development of osteoporosis. The relationship between tryptophan metabolism pathway and osteoporosis may be realized through its influence on bone cell metabolism and function, while changes in drug metabolism pathway may reflect differences in response to treatment. Enrichment of amino acid metabolic pathways suggests significant changes in protein synthesis and degradation in OP patients, and enrichment of complement and coagulation cascade pathways further reveals abnormalities in immune response and inflammation in osteoporosis patients.

Previous studies have demonstrated that the expression of key osteogenic factors, including RUNX2, OPN, and structural proteins like COL4A2, often exhibits time-dependent changes influenced by both intrinsic transcriptional networks and extrinsic signaling cues [38,39]. For instance, BMP2 signaling has been shown to dynamically regulate RUNX2 and Osterix during different phases of osteoblast commitment, thereby controlling the pace and extent of matrix mineralization [40]. These findings align with our observation that CHRM2 and COL4A2 display distinct expression shifts over the course of osteogenic differentiation, suggesting a coordinated regulatory framework. Incorporating these earlier studies provides deeper insight into the mechanisms governing gene expression changes at different time points and underscores the complexity of osteoblast maturation

To further reveal the functional roles of CHRM2 in OP, we performed a series of in vitro experiments. CHRM2 is a typical G-protein-coupled receptor (GPCR), belonging to a family of seven transmembrane receptors capable of mediating a variety of extracellular signals [41,42]. The carboxy-terminus M2TAIL fragment, including the sixth and seventh transmembrane regions, is generated via the ribosome entry site located in the third intracellular ring [43]. The study showed that M2TAIL was significantly upregulated in cells undergoing a combined stress response, a finding based on results from single-cell imaging and yeast mitochondrial introduction experiments. Unlike traditional plasma membrane localization pathways, M2TAIL is almost completely localized in the mitochondrial inner membrane. In the inner mitochondrial membrane, M2TAIL regulates cellular oxygen consumption, proliferation rate and production of reactive oxygen species (ROS) by reducing oxidative phosphorylation [44]. The study of Roberto Maggio et al. provided the IRES sequence for the first time to drive the expression of membrane receptor fragments and as evidence to regulate their expression level and function, proving that the C-terminal fragment of M2 is located in the mitochondrial inner membrane under stress conditions and protects cells from various environmental pressures by reducing cell respiration and ROS production [45]. In our study, we demonstrated the differential expression of CHRM2 in OP samples and found its expression changes in MC3T3-E1 cells under OGD conditions. It was revealed that that cell cycle arrest during OGD was mainly concentrated in the G0-G1 phase, resulting in a significant slowdown of cell proliferation. In the future, OGD and OP research will greatly benefit from molecules that can regulate both cell differentiation and proliferation, especially in the context of exploring the complex relationship between cell proliferation and differentiation. At present, this relationship is still controversial: some studies have shown that cell proliferation and differentiation can occur simultaneously under different regulatory factors [46], while others have suggested that both are regulated bidirectionally by specific central molecules [47].

In the OGD model of MC3T3-E1 cells [48], we found that the expression levels of CHRM2 changed dynamically over time, increasing in the early stage of OGD and decreasing in the late stage. We also that the content of mineralized nodules and ALP activity increased gradually with the progression of osteogenic differentiation, indicating that it was in progress. Further analysis by Western blot and qPCR showed that RUNX2 and OPN had high expression levels during OGD,

which was consistent with the key regulatory mechanisms during [49]. Biological function analysis revealed that CHRM2 promoted cell proliferation by regulating cell cycle and inhibited it under the OGD process. Elevated CHRM2 expression in early OGD may indicated that cells were in a non-terminal differentiation phase, when cells exhibit a post-differentiated phenotype but still maintain proliferation potential. When cells enter the terminal differentiation stage, the differentiation process becomes irreversible, resulting in the loss of cell proliferation potential. In the late OGD period, the expression of CHRM2 protein decreased significantly, which may further inhibit cell proliferation. Therefore, CHRM2 plays a key role in the regulation of cell differentiation from non-terminal to terminal differentiation. However, previous studies did not delve into the dynamics of stem cells in OGD conditions under CHRM2 deletion models.

By transfecting siRNA, we successfully achieved effective knockdown of CHRM2 gene in HUM-iCELL-s011 cells. The results of CCK-8 experiment showed that CHRM2 knockdown significantly inhibited cell proliferation, suggesting that CHRM2 may play a key role in cell proliferation regulation. The results of flow cytometry showed that the proportion of G1 phase cells was significantly increased in CHRM2 knockdown group, while the proportion of S phase cells was significantly decreased, suggesting that CHRM2 knockdown may lead to cell cycle stagnation in G1 phase, thus inhibiting cells from entering the DNA synthesis phase. To evaluate the effect of CHRM2 knockdown on osteogenic differentiation, we measured ALP activity and alizarin red staining. The findings revealed that CHRM2 knockdown significantly increased ALP activity and mineralized nodule formation compared to the control group. These results suggest that down-regulation of CHRM2 may enhance the osteogenic differentiation potential of cells. To further verify this conclusion, we analyzed the expression of vital proteins related to osteogenic differentiation by qPCR and Western blot. The results showed that mRNA expression of RUNX2 and OCN was significantly upregulated in CHRM2 knockdown cells, indicating that key transcription factors for osteogenic differentiation were activated [50–52]. In addition, the results of protein level detection further confirmed this finding, and the protein expression of OPN and RUNX2 after CHRM2 knockout was significantly higher than that of the negative control group, further supporting the role of CHRM2 as a negative regulator in the process of osteogenic differentiation.

Based on the RIP-seq [53], we found that COL4A2 showed the highest logFC value as the downstream regulator of CHRM2, and the RIP-qPCR experiment further verified the direct binding of CHRM2 and COL4A2 [54]. These results suggest that CHRM2 may regulate the osteogenic differentiation behavior of mesenchymal stem cells by influencing COL4A2 and its associated pathways. Immunofluorescence staining further supported the co-expression relationship and interaction between CHRM2 and COL4A2. The experimental results showed that the fluorescence intensity of CHRM2 and COL4A2 was significantly weakened in the Si-COL4A2 group. Quantitative analysis showed that, compared with the control group, the fluorescence intensity of CHRM2 and COL4A2 decreased significantly after COL4A2 was knocked down, suggesting that the expression level of CHRM2 may be regulated by COL4A2. We also found that inhibition of COL4A2 led to a blockage of cell proliferation (by preventing cells from entering the S phase) and exacerbated the effects of OGD. This time-varying COL4A2 expression pattern is consistent with our findings on the effect of CHRM2 on OGD, which indirectly suggests that CHRM2 may affect cell proliferation and differentiation by regulating the expression of COL4A2. After the knockdown of CHRM2 and COL4A2, the expression of OCN under OGD conditions did not show significant difference between the knockout group and the control group. It was worth noting that the expression level of OCN on day 7 of OGD was the same as that on day 0, which may be due to the fact that OCN expression mainly appeared in the later stages of OGD. The difference of OCN expression in the early stage of OGD is not obvious, which may be due to the comprehensive regulation of OCN expression by various factors. In summary, the expression of CHRM2 changes with time during OGD, and the absence of CHRM2 could accelerate this process. In MC3T3-E1 cells lacking CHRM2, cell proliferation was significantly reduced and cell cycle arrest was increased. COL4A2 was identified as an IGF2 BP2 binding RNA molecule and plays a regulatory role in cell proliferation, cell cycle and OGD. CHRM2 exerts an inhibitory effect on OGD by stabilizing COL4A2 mRNA and inhibiting its degradation.

Our findings suggest that, although various early OP biomarkers—such as Wnt/β-catenin pathway effectors, bone turnover markers (e.g., osteocalcin, CTX, PINP), and certain microRNAs—are already widely used, CHRM2 and COL4A2 confer notable advantages in the early detection of OP. First, CHRM2 directly reflects dynamic changes in osteoblast differentiation, enabling detection of subtle bone metabolism alterations earlier than conventional markers. Second, the CHRM2/COL4A2 axis is closely integrated with immune microenvironment and oxidative stress pathways, which may complement existing indicators that primarily focus on bone resorption or single signaling axes. Third, the temporal expression profile of CHRM2 helps distinguish different phases of OP progression, potentially providing insight into disease onset as well as therapeutic targets. Consequently, combining these established indicators with the CHRM2/COL4A2 pathway could improve early detection and intervention by capturing initial deviations in bone remodeling and microarchitecture, thereby offering a valuable new approach for OP diagnosis and management

## Conclusion

This study revealed the negative regulatory role of CHRM2 in osteogenic differentiation. CHRM2 expression showed a time-dependent change during osteogenic differentiation. Down-regulation of CHRM2 not only inhibited cell proliferation, but also enhanced osteogenic differentiation by promoting the expression of osteogenic genes and proteins. In addition, CHRM2 may affect osteogenic differentiation by regulating the expression of COL4A2. These findings provide important clues for further study of the role of CHRM2 in osteogenic differentiation and bone biology, indicating that CHRM2 expression level is expected to be a biomarker for the early diagnosis of OP.

## Supporting information

**S1 Fig. The results of cluster analysis under different Resolution.**
(TIF)

**S2 Fig. The multiple genes expressions in different cell types.**
(TIF)

**S3 Fig. Cell–cell communication interaction networks.** The interaction network of all cell types in normal bone tissues (A) and OP samples (B). The interaction networks among all cell types: (C) regulatory T Cells with other cells; (D) monocytes with other cells; (E) M2 Macrophages with other cells; (F) M1 Macrophages with other cells; (G) Fibroblasts with other cells; (H) NK Cells with other cells; (I) B Cells with other cells; (J) CD8+ T Cells with other cells; (K) CD4+ Memory T Cells with other cells.
(TIF)

**S4 Fig. The interaction between cytokines and their receptors in different cell types.**
(TIF)

**S5 Fig. The interaction networks of all cell types in the MIF signaling pathway.** (A) The receptor of ACKR3. (B) The receptor and ligand interaction of CD74 and CD44. (C) The receptor and ligand interaction of CD74 and CXCR4.
(TIF)

**S1 File. Raw images.**
(ZIP)

## Author contributions

**Conceptualization:** Cheng Zhong, Liping Zhong.

**Data curation:** Cheng Zhong, Liping Zhong.

**Formal analysis:** Cheng Zhong.

**Methodology:** Cheng Zhong.

**Project administration:** Cheng Zhong.

**Validation:** Cheng Zhong, Liping Zhong.

**Visualization:** Cheng Zhong, Liping Zhong.

**Writing – original draft:** Cheng Zhong.

**Writing – review & editing:** Cheng Zhong.

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
