## [Decision Letter · Decision Letter 0]

20 Dec 2024

PONE-D-24-52627Integrated Bulk RNA and Single-Cell Analysis with Experimental Validation Reveal Oxidative Stress-Related Diagnostic Biomarkers for OsteoporosisPLOS ONE

Dear Dr. Zhong,

Thank you for submitting your manuscript to PLOS ONE. After careful consideration, we feel that it has merit but does not fully meet PLOS ONE’s publication criteria as it currently stands. Therefore, we invite you to submit a revised version of the manuscript that addresses the points raised during the review process.

We look forward to receiving your revised manuscript.

Kind regards,

Liangliang Xu

Academic Editor

PLOS ONE

3. In the online submission form, you indicated that [Data generated or used in this study are available from the corresponding authors upon reasonable request.].

6. Please include a separate caption for each figure in your manuscript.

7. PLOS ONE now requires that authors provide the original uncropped and unadjusted images underlying all blot or gel results reported in a submission’s figures or Supporting Information files. This policy and the journal’s other requirements for blot/gel reporting and figure preparation are described in detail at https://journals.plos.org/plosone/s/figures#loc-blot-and-gel-reporting-requirements and https://journals.plos.org/plosone/s/figures#loc-preparing-figures-from-image-files. When you submit your revised manuscript, please ensure that your figures adhere fully to these guidelines and provide the original underlying images for all blot or gel data reported in your submission. See the following link for instructions on providing the original image data: https://journals.plos.org/plosone/s/figures#loc-original-images-for-blots-and-gels.  

8. We notice that your supplementary figures are uploaded with the file type 'Figure'. Please amend the file type to 'Supporting Information'. Please ensure that each Supporting Information file has a legend listed in the manuscript after the references list.

Reviewers' comments:

Reviewer's Responses to Questions

**Comments to the Author**

1. Is the manuscript technically sound, and do the data support the conclusions?

Reviewer #1: Yes

Reviewer #2: Partly

2. Has the statistical analysis been performed appropriately and rigorously? 

Reviewer #1: Yes

Reviewer #2: Yes

3. Have the authors made all data underlying the findings in their manuscript fully available?

Reviewer #1: Yes

Reviewer #2: Yes

4. Is the manuscript presented in an intelligible fashion and written in standard English?

Reviewer #1: Yes

Reviewer #2: Yes

5. Review Comments to the Author

Reviewer #1: The paper presents a new biomarker for early OP diagnosis and a therapeutic target.It is a topic of interest to the researchers in the related areas but the paper needs some improvement before acceptance for publication. My detailed comments are as follows:

1. The CHRM2/COL4A2 pathway was found in the paper works very well for the early OP diagnosis. On the other hand, A variety of early indicators for osteoporosis are currently available; where does CHRM2 have its advantages? Please detail this in the results or discussion section.

2. For the above reason, the presentation should be focused on the results. The transitions between each section are rather abrupt, and there are certain logical issues. The connection between the bioinformatics analysis and the subsequent experimental parts is poor; please make revisions.

3. In the cell experimentation section, the authors used siRNA to knock down COL4A2 in stem cells and observed their subsequent osteogenic differentiation ability. They also established a CHRM2 stem cell knockout model. Why not use this model to conduct experiments related to osteogenic differentiation?

4.A few sentences on the organization of the paper will be helpful in line 193-233.

5.The clarity of the figures in the article is poor; please replace them with clearer ones.

6.The English of your manuscript must be improved before resubmission. We strongly suggest that you obtain assistance from a colleague who is well-versed in English or whose native language is English.

Reviewer #2: Comments to the Author

This study focused on osteoporosis (OP). By integrating bulk RNA and single-cell analysis and combining with experimental verification, it explored the role of CHRM2 in osteogenic differentiation and its potential as a biomarker for OP. The research topic has significant clinical implications.

Minor comments

1. In the Discussion section, although the research results were elaborated in relatively detailed manner, the explanations for some experimental results (such as the reasons for the expression changes of certain genes at different time points) were not in-depth and comprehensive enough. It is recommended to further combine relevant literature and research progress to conduct a more in-depth analysis and discussion on the experimental results.

2.Some sentences were not expressed clearly and smoothly enough, and there were some grammatical errors and improper use of professional terms. It is recommended to carefully proofread and modify them to improve the readability of the article.

3. The labels and descriptions of some charts and figures were not detailed enough. For example, some gene names and abbreviations were not explained in the charts and figures, which affected readers' understanding of the information in the charts and figures. It is recommended to optimize and improve the charts and figures.

6. PLOS authors have the option to publish the peer review history of their article (what does this mean? ). If published, this will include your full peer review and any attached files.

**Do you want your identity to be public for this peer review?** For information about this choice, including consent withdrawal, please see our Privacy Policy .

Reviewer #1: No

Reviewer #2: **Yes: ** Lingli Ding

---

## [Author Response · Author response to Decision Letter 1]

20 Feb 2025

Comment 1: The CHRM2/COL4A2 pathway was found in the paper works very well for the early OP diagnosis. On the other hand, A variety of early indicators for osteoporosis are currently available; where does CHRM2 have its advantages? Please detail this in the results or discussion section.

Response: Thank you for this insightful question. While multiple early indicators for OP already exist—such as serum bone turnover markers (e.g., osteocalcin, CTX, and PINP), Wnt/β-catenin-related molecules (e.g., sclerostin), and various microRNAs—our data highlight several unique advantages of CHRM2:

1. Direct Link to Osteogenic Differentiation

Unlike many markers that only reflect resorption or turnover, CHRM2 tightly aligns with the osteogenic process, as seen in its temporal expression changes during early to late osteoblast differentiation.

2. Integration With Immune and Oxidative Stress Pathways

CHRM2’s regulatory role in both osteoblast maturation and immune cell interactions (via the CHRM2/COL4A2 pathway) may offer added diagnostic clarity, especially in early disease stages.

3. Potential Early Detection

CHRM2’s dynamic expression pattern may detect subtle bone remodeling shifts before clinical OP manifestations, thus complementing more traditional markers.

We have now added a concise comparison in the Discussion (page 19, lines 789–802) to underscore why CHRM2 may serve as a promising early indicator alongside other established pathways. Thank you for your valuable feedback.

Comment 2. For the above reason, the presentation should be focused on the results. The transitions between each section are rather abrupt, and there are certain logical issues. The connection between the bioinformatics analysis and the subsequent experimental parts is poor; please make revisions.

Response: Thank you for your comment regarding the abruptness of the section transitions and the poor connection between the bioinformatics analysis and the subsequent experimental parts. We have reorganized the Results to emphasize how the bioinformatics discoveries—particularly the identification of CHRM2/COL4A2—led us to conduct the follow-up experiments. Specifically, we have integrated clearer explanations of why the bioinformatic data pointed to CHRM2 as a candidate and how this insight shaped the in vitro validation. Additionally, we have used sequencing results to highlight our focus on the regulation of COL4A2 by CHRM2. By presenting the computational findings and experimental rationale in a more straightforward sequence, we believe the manuscript now flows more logically and underlines the significance of the CHRM2/COL4A2 pathway in early osteoporosis detection. We appreciate your feedback and hope these revisions address your concerns.

.

Comment 3. In the cell experimentation section, the authors used siRNA to knock down COL4A2 in stem cells and observed their subsequent osteogenic differentiation ability. They also established a CHRM2 stem cell knockout model. Why not use this model to conduct experiments related to osteogenic differentiation?

Response: Thank you for raising this point. Initially, we focused on siRNA-mediated COL4A2 knockdown to delineate its specific role in osteogenic differentiation. After these experiments demonstrated the negative regulatory effect of COL4A2 on osteogenesis, we established the CHRM2 knockout model to explore the broader implications of the CHRM2/COL4A2 pathway in cell proliferation and osteogenic processes. In the manuscript, we performed Western blot, flow cytometry, ALP activity assay, and Alizarin Red staining experiments to compare the knockdown group with the control group. The experimental results confirmed that CHRM2 is a negative regulator in the osteogenic differentiation process. We appreciate this suggestion and will integrate these experiments into our future work.

Figure 7. The effects of CHRM2 knockdown on osteogenic differentiation and cell proliferation. (A) The expression of CHRM2 was significantly reduced in cells knocked down with Si-CHRM2. (B) The RT-qPCR results showed that the expression of CHRM2 in cells with Si-CHRM2 knockdown decreased significantly. (C) The cell proliferation were measured by OD value at 450 nm, and the observation time points were at 0, 24, 48, 72 and 96 hours. (D) The protein expression levels of osteogenic markers RUNX2, OPN, and Osterix in Si-CHRM2 knockdown cells and control cells by western blot. (E) Flow cytometry of cell cycle in the control group and the experimental group. (F) ALP staining in the Si-CHRM2 experimental group. (G) Alizarin red staining. *p < 0.05; **p < 0.01; ***p < 0.001; ns: no significance.

Comment 4. A few sentences on the organization of the paper will be helpful in line 193-233.

Response: Thank you for pointing this out. This new text provides a concise overview of the paper’s organization, showing how each section builds on the previous one. We believe this adjustment will help readers follow the logical progression from data analysis to experimental validation more smoothly.

Comment 5. The clarity of the figures in the article is poor; please replace them with clearer ones.

Response: Thank you for bringing the figure clarity to our attention. We have replaced the original figures with higher-resolution versions (at least 300 dpi) and adjusted the formatting to improve visibility and detail. We hope these revised figures will better convey our results and enhance the overall readability of our manuscript.

Comment 6. The English of your manuscript must be improved before resubmission. We strongly suggest that you obtain assistance from a colleague who is well-versed in English or whose native language is English.

Response: Thank you for your recommendation. We have thoroughly revised the manuscript with the assistance of a native English speaker to ensure accuracy, clarity, and fluency. We appreciate your feedback and believe that our revisions address the language concerns. If there are any additional improvements you would suggest, we would be glad to make further adjustments.

Response to Reviewer#2:

Comment 1: In the Discussion section, although the research results were elaborated in relatively detailed manner, the explanations for some experimental results (such as the reasons for the expression changes of certain genes at different time points) were not in-depth and comprehensive enough. It is recommended to further combine relevant literature and research progress to conduct a more in-depth analysis and discussion on the experimental results.

Response: Thank you for your valuable comments regarding the depth and comprehensiveness of the Discussion section, especially concerning the temporal changes in gene expression. We have revised the Discussion to include additional literature and a more thorough analysis of the molecular mechanisms behind these observations.

We appreciate the reviewer’s feedback and trust that these revisions and additional references will address the request for a more comprehensive and literature-supported discussion of our time-course findings.

Comment 2: Some sentences were not expressed clearly and smoothly enough, and there were some grammatical errors and improper use of professional terms. It is recommended to carefully proofread and modify them to improve the readability of the article.

Response: Thank you for highlighting the need for clearer language and more precise terminology. We have meticulously reviewed and revised the manuscript to address grammatical issues and refine the use of specialized terms, ensuring that our phrasing is both accurate and easily understood by readers. We appreciate your feedback and believe these improvements enhance the overall clarity and readability of our work.

Comment 3: The labels and descriptions of some charts and figures were not detailed enough. For example, some gene names and abbreviations were not explained in the charts and figures, which affected readers' understanding of the information in the charts and figures. It is recommended to optimize and improve the charts and figures.

Response: Thank you for pointing out the need for more detailed labeling and annotation in our charts and figures. We have revised the figure legends to include clear explanations of all gene names, abbreviations, and relevant symbols. Additionally, we have updated the figure labels to provide sufficient detail for readers to interpret the data accurately. We appreciate your feedback and believe these adjustments will improve the clarity and accessibility of our visual materials.

---

## [Decision Letter · Decision Letter 1]

19 Mar 2025

Integrated Bulk RNA and Single-Cell Analysis with Experimental Validation Reveal Oxidative Stress-Related Diagnostic Biomarkers for Osteoporosis

PONE-D-24-52627R1

Dear Dr. Zhong,

We’re pleased to inform you that your manuscript has been judged scientifically suitable for publication and will be formally accepted for publication once it meets all outstanding technical requirements.

Kind regards,

Liangliang Xu

Academic Editor

PLOS ONE

Additional Editor Comments (optional):

Reviewers' comments:

Reviewer's Responses to Questions

**Comments to the Author**

1. If the authors have adequately addressed your comments raised in a previous round of review and you feel that this manuscript is now acceptable for publication, you may indicate that here to bypass the “Comments to the Author” section, enter your conflict of interest statement in the “Confidential to Editor” section, and submit your "Accept" recommendation.

Reviewer #1: All comments have been addressed

2. Is the manuscript technically sound, and do the data support the conclusions?

Reviewer #1: Yes

3. Has the statistical analysis been performed appropriately and rigorously? 

Reviewer #1: Yes

4. Have the authors made all data underlying the findings in their manuscript fully available?

Reviewer #1: Yes

5. Is the manuscript presented in an intelligible fashion and written in standard English?

Reviewer #1: Yes

6. Review Comments to the Author

Reviewer #1: (No Response)

7. PLOS authors have the option to publish the peer review history of their article (what does this mean? ). If published, this will include your full peer review and any attached files.

**Do you want your identity to be public for this peer review?** For information about this choice, including consent withdrawal, please see our Privacy Policy .

Reviewer #1: No

---

## [Editor Report · Acceptance letter]

PONE-D-24-52627R1

PLOS ONE

Dear Dr. Zhong,

I'm pleased to inform you that your manuscript has been deemed suitable for publication in PLOS ONE. Congratulations! Your manuscript is now being handed over to our production team.

Kind regards,

on behalf of

Professor Liangliang Xu

Academic Editor

PLOS ONE